# Multi-Dimensional Knowledge Profiling with Large-Scale Literature Database and Hierarchical Retrieval

## Abstract

The rapid expansion of research across machine learning, vision, and language has produced a volume of publications that is increasingly difficult to synthesize. Traditional bibliometric tools rely mainly on metadata and offer limited visibility into the semantic content of papers, making it hard to track how research themes evolve over time or how different areas influence one another. To obtain a clearer picture of recent developments, we compile a unified corpus of more than 100,000 papers from 22 major conferences between 2020 and 2025 and construct a multidimensional profiling pipeline to organize and analyze their textual content. By combining topic clustering, LLM-assisted parsing, and structured retrieval, we derive a comprehensive representation of research activity that supports the study of topic lifecycles, methodological transitions, dataset and model usage patterns, and institutional research directions. Our analysis highlights several notable shifts, including the growth of safety, multimodal reasoning, and agent-oriented studies, as well as the gradual stabilization of areas such as neural machine translation and graph-based methods. These findings provide an evidence-based view of how AI research is evolving and offer a resource for understanding broader trends and identifying emerging directions.

## 1 Introduction

The scale and diversity of contemporary AI research continue to grow at an extraordinary pace. Across computer vision, machine learning, natural language processing, and related areas, the past five years have seen rapid shifts in model architectures, training strategies, datasets, benchmarks, and application domains. For researchers, this expansion brings substantial challenges: it is increasingly difficult to situate individual works within broader developments, to track how research themes evolve, or to identify areas that are emerging, stabilizing, or declining.

Existing tools only partially address this need. Traditional bibliometric methods, which built on metadata, co-citation networks, and keyword statistics Zupic & Čater (2015); Donthu et al. (2021); Grootendorst (2022), provide high-level overviews but capture limited semantic information and treat topics as largely static entities. As a result, they offer only coarse insight into methodological transitions, cross-domain influences, or the finer-grained structure of research problems. Meanwhile, recent systems that incorporate large language models (LLMs) Brown et al. (2020); Achiam et al. (2023) demonstrate improved semantic analysis, supporting tasks such as retrieval-augmented question answering Gao et al. (2023) and automated survey generation Wang et al. (2024b). However, these tools are typically designed for short-range retrieval, single papers, or narrow tasks, and they do not provide a coherent, longitudinal view of large scientific corpora.

These gaps point to the need for a unified way to organize, summarize, and interpret the rapidly expanding body of AI literature. In this work, we construct a large-scale profiling pipeline aimed at characterizing the recent development of AI research. Using more than 100,000 papers from 22 major conferences published between 2020 and 2025 (Fig. 1), we combine text clustering, LLM-assisted semantic parsing, and lightweight retrieval techniques to form a structured representation of research problems, methods, datasets, and topical

dynamics. Rather than emphasizing algorithmic novelty, our focus is on creating a coherent analytic framework that enables researchers to explore and reason about the field at multiple levels of granularity.

Our study provides two complementary benefits. First, we derive a high-resolution view of topic lifecycles, dataset and model adoption patterns, and methodological transitions across areas such as vision, multimodal learning, foundation models, and generative modeling. Second, by incorporating structured retrieval and semantic filtering, we enable grounded, evidence-based queries that support practical research tasks, such as surveying subfields, tracing the evolution of techniques, or identifying emerging directions.

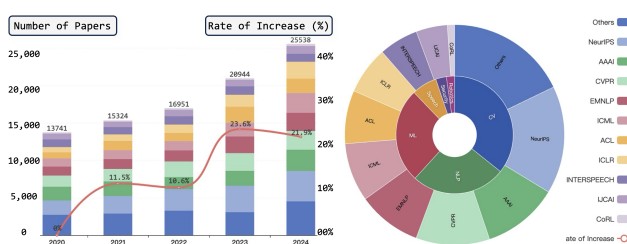

Figure 1: Number of papers published and topic statistics across 22 conferences from 2020 to 2025.

Through this analysis, we highlight several notable shifts in the AI landscape, including the consolidation of previously fast-moving areas, the rise of multimodal and agent-oriented research, and clear transitions in compute usage and model scaling practices. We expect the resulting knowledge database to serve as a resource for understanding broad trends, informing future meta-analyses, and supporting data-driven research planning.

Our work makes the following contributions:

- We construct a large-scale profiling pipeline over more than 100,000 papers from 22 major conferences, enabling structured analysis of semantic topics, methods, datasets, and research trajectories.

- We integrate clustering-based topic organization with LLM-assisted parsing and retrieval, producing a structured and interpretable representation of the AI research landscape that complements traditional bibliometrics.

- We conduct comprehensive empirical analyses, revealing trends in topic evolution, emerging subfields, methodological transitions, dataset and model dynamics, and institutional research patterns across the broader AI community.

Overall, these results provide an evidence-based view of how modern AI research is evolving and offer a foundation for transparent, large-scale, and semantically grounded scientometric analysis.

## 2 Related Work

### 2.1 Bibliometrics & Scientific Trend Analysis

Content mining and trend analysis of scientific literature are important topics in scientometrics and information science. Traditional bibliometric approaches rely on metadata, extracting abstracts, authors, journals, keywords, and citationsZupic & Čater (2015); Donthu et al. (2021), and applying co-citation and co-word analysesVan Eck & Waltman (2010); Chen (2006); Aria & Cuccurullo (2017) to reveal the structure and evolution of research fields. However, these methods mainly rely on surface features and are limited in capturing the latent semantic structures within documents. To address this, topic modeling and unsupervised clustering have been widely usedBlei (2012), with LDA inferring topic distributions from word co-occurrencesBlei et al. (2003); AlSumait et al. (2009); Chang et al. (2009); Chuang et al. (2013). More recently, embedding-based clustering methods, such as BERTopicGrootendorst (2022); Reimers & Gurevych (2019) combined with BERTDevlin et al. (2019), and applications of large language modelsKostikova et al. (2025); Diaz-Rodriguez (2025); Çelikten & Onan (2025); Lam et al. (2024) have enhanced the discovery of latent topics. Nevertheless, existing methods still struggle to systematically represent the dynamic evolution of knowledgeSi et al. (2024); Scherbakov et al. (2025), and a unified framework for multidimensional knowledge profiling remains lacking.

## 2.2 LLMs for Document Understanding

Trained on massive general-domain corpora, LLMs have shown strong abilities in semantic understanding and text generation Brown et al. (2020), giving rise to series of GPT Achiam et al. (2023) and other state-of-the-art models Xie et al. (2026); Comanici et al. (2025); Touvron et al. (2023); Yang et al. (2025). To support scientific applications, researchers have explored LLMs in research workflows. For example, PaperQA Lála et al. (2023); Skarlinski et al. (2024); Besrour et al. (2025) leverages retrieval-augmented generation (RAG) techniques Xue et al. (2025); Gao et al. (2023); Li et al. (2025); Cheng et al. (2025) to enable document question answering, while multi-agent and reinforcement learning approaches Nguyen et al. (2025); Singh et al. (2025); Li et al. (2025); Wu et al. (2025) further improve system accuracy. Another line of work focuses on automating literature review generation, including AutoSurvey Wang et al. (2024b), SurveyXLiang et al. (2025), SurveyForge Liang et al. (2025); Yan et al. (2025), and SurveyG Nguye et al. (2025), which implement pipelines for retrieval, filtering, organization, and writing. Despite these advances, existing LLM-based approaches are generally limited in scope: they focus on single tasks, rely on predefined corpora, and seldom provide a unified view of scientific knowledge evolution across multiple dimensions, including method evolution, task evolution, dataset adoption, and compute trends. Moreover, few approaches systematically integrate temporal information, semantic structure, and cross-conference knowledge, limiting their utility for large-scale trend analysis and evidence-based research decision making.

In this work, we propose a multidimensional profiling framework that leverages LLMs to extract and organize semantic information from a large corpus of scientific publications. By combining embeddings, topic clustering, and hierarchical retrieval, the framework produces a dynamic, hierarchical view of knowledge evolution, enabling trend analysis, cross-domain comparisons, and fine-grained topic investigation, and supporting data-driven exploration and research planning.

## 3 Knowledge Profiling Framework

Table 1: Detailed annual paper distribution for the 22 major venues in *ResearchDB* from 2020 to 2025.

| Venue Group | 2020 | 2021 | 2022 | 2023 | 2024 | 2025 | Total |
|---|---|---|---|---|---|---|---|
| Core ML/Theory (NeurIPS, ICML, ICLR, UAI, COLT) | 264 | 4703 | 5587 | 7317 | 9596 | 7269 | 34736 |
| Computer Vision (CVPR, ICCV, ECCV) | 1358 | 3272 | 3715 | 4509 | 4868 | 2871 | 20593 |
| NLP/Speech (ACL, EMNLP, NAACL, COLM, INTERSPEECH, IWSLT) | 3098 | 4138 | 4498 | 5580 | 6417 | 4730 | 28461 |
| General/Applied AI (AAAI, IJCAI, CoRL) | 2644 | 2837 | 2683 | 3071 | 4178 | 3239 | 18652 |
| Systems/Security (MLSYS, OSDI, NDSS, etc.) | 216 | 374 | 468 | 467 | 479 | 36 | 2040 |
| **Overall Total (22 Venues)** | **7580** | **15324** | **16951** | **20944** | **25538** | **18145** | **104482** |

We collected papers accepted between 2020 and 2025 from 22 academic conferences, including CVPR, ECCV, ICCV, ICLR, NeurIPS, and ICML, along with the citation counts for each paper, and over 100,000 papers are included in total, as shown in Tab. 1. Our main focus is on how to efficiently perform Multi-Dimensional Knowledge Profiling on this scientific data, as illustrated in Fig. 2.

### 3.1 Semantic Parsing & Knowledge Extraction

To systematically analyze the rapidly growing body of AI research literature, we propose a multi-stage knowledge extraction framework that combines automated content mining with LLM-driven semantic analysis. First, considering the difficulties in structured output and efficiency when directly using LLMs to analyze PDF files, we employ minerU Wang et al. (2024a) to convert the collected PDFs into structured Markdown files. Subsequently, based on the Markdown file corresponding to each paper, we utilize Deepseek-R1-32B Guo et al. (2025) to conduct multi-dimensional analysis of the papers, including meta information, abstract, research questions, methods, datasets, evaluation metrics, model architecture, contributions, limitations, and future directions.

Topic information is a key component of multi-dimensional content parsing. We use clustering not primarily to reduce LLM cost, but to impose a corpus-level topic structure before topic naming. Direct per-paper LLM classification can be useful for closed-label settings, but in an open and rapidly changing corpus it may

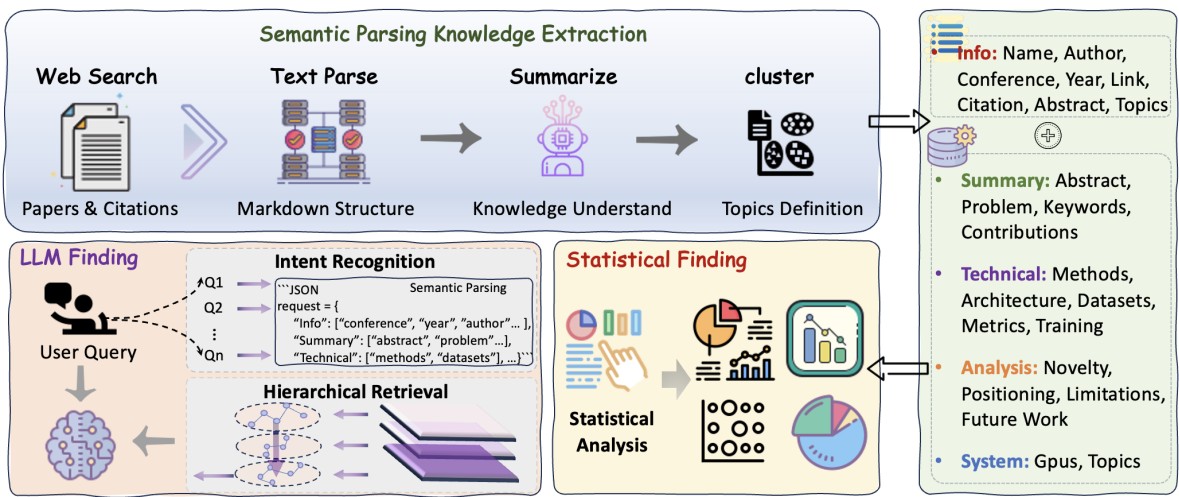

Figure 2: LLM-Driven Multi-Dimensional Knowledge Profiling Framework with Large-Scale Database and Hierarchical Retrieval.

introduce inconsistent topic names and drifting granularities across papers and years. We therefore adopt a BERTopic-style pipelineGrootendorst (2022): a text encoder embeds the title and abstract of each paper, UMAPMcInnes et al. (2018) reduces the dimensionality of the embeddings, and HDBSCANMcInnes et al. (2017) groups papers into more than 300 semantically coherent topic categories. ChatGPT-5 is then used only after clustering to summarize each cluster, assign readable topic names, and refine the hierarchical topic relationships. This design makes the topic labels depend on corpus-level semantic neighborhoods while still benefiting from LLM-based naming.

The ResearchDB schema is organized into five broad categories rather than standalone fields: Info, Summary, Technical, Analysis, and System. These categories match the framework overview in Fig. 2, while fields derived after extraction are assigned to the closest functional category, such as topic labels under Info and trend-level annotations under Analysis. Tab. 2 summarizes the main field groups in the main text, and Appendix A provides the complete schema, prompts, and a worked example for reproducibility. Unlike traditional bibliometric methods, this framework relies on semantic understanding rather than mere keywords, enabling the discovery of emerging or previously underexplored research topics, and finally obtaining the following information:

Table 2: Main field groups in ResearchDB. The full field-level schema is provided in Appendix A.

| Field Group | Main Fields | Purpose/Description |
|---|---|---|
| Info | paper_name, authors, conference, year, institution, topic_name/topic_ID | Records bibliographic metadata, affiliation information, and derived topic labels for filtering, grouping, and corpus-level statistics. |
| Summary | abstract_ori, abstract_summary, keywords, keywords_description, problem_statement, contributions | Captures the paper's core problem, abstract-level content, keywords, and main contributions. |
| Technical | methods, architecture, loss_function, datasets, metrics, datasets_metrics_mapping | Represents method design, model structure, objectives, datasets, and evaluation settings for methodological and benchmark analysis. |
| Analysis | experiments, results_summary, limitations, future_work, novelty_type, field_positioning, trend_insight | Summarizes empirical evidence, limitations, future directions, novelty type, and trend-level interpretation. |
| System | training_setup, gpu_info | Records resource-related training configuration and explicitly reported compute information for scaling and compute-trend analysis. |

These field groups provide the structured representation used for both aggregate trend analysis and the retrieval pipeline described below.

## 3.2 Intent-Driven Knowledge Retrieval

Large language models have shown potential for literature retrieval and summarization, but direct application in research workflows faces key limitations: hallucination, omission or misinterpretation of complex information, and poor grounding in evidence, particularly over large, multi-year corpora. To address these challenges, we develop an intent-driven hierarchical retrieval pipeline over our structured *ResearchDB*, combining metadata filtering with weighted multi-field semantic search to provide reliable, evidence-based input to language models.

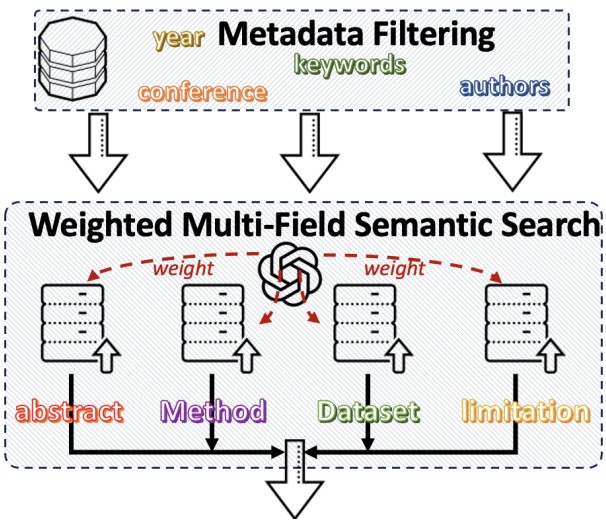

Figure 3: The Framework of Hierarchical Retrieval.

**Intent Recognition** To handle complex queries $Q$, we first decompose $Q$ into simpler sub-questions $\{Q_i\}$ that isolate distinct information needs. Query decomposition operates at the task level: for example, a request about "dataset changes in multi-modal reasoning from 2021 to 2025" may be split into dataset, topic, and temporal sub-questions. Each sub-question is then parsed into structured retrieval instructions in JSON format, specifying relevant metadata constraints, keywords, entities, and content types. This decomposition reduces ambiguity, ensures comprehensive coverage, and provides clear guidance for the hierarchical retrieval stage.

**Hierarchical Retrieval** Each sub-question $Q_i$ is addressed through a two-level retrieval strategy, as shown in Fig. 3.

1. **Metadata Filtering.** We filter *ResearchDB* using metadata attributes such as conference, year, authors, and keywords to obtain a candidate subset. Among them, keywords use fuzzy matching, and other fields use exact matching.

$$D_{\text{filtered}} = \text{Filter}(ResearchDB, \text{metadata}). \tag{1}$$

2. **Weighted Multi-Field Semantic Search.**

   For the document subset $D_{\text{filtered}}$ obtained via metadata filtering, we perform a subsequent intent-guided semantic search. We implement this step as *rule-prior-initialized, intent-guided adaptive field weighting.* The mechanism is neither a learned weighting model nor an unconstrained weight generator. Instead, the LLM first identifies the intent of each sub-question and selects the most relevant candidate fields from the *ResearchDB* schema; then the retrieval module initializes field weights using intuitive prior weights summarized from typical query patterns and pilot retrieval experiments. For instance, dataset or benchmark queries emphasize `datasets` and `metrics`, method-evolution queries emphasize `methods` and `architecture`, and compute-related queries emphasize `gpu_info` and `training_setup`. Because different queries activate different candidate fields, the resulting field-weight configuration is query-adaptive while remaining transparent and reproducible. These weights are applied to semantic similarity scores $s_{ij}$ computed over the selected searchable fields. The final weighted relevance score $S_i$ is computed as:

$$S_i = \sum_{j=1}^{n} w_j \cdot s_{ij}, \quad \text{with} \quad \sum_{j=1}^{n} w_j = 1. \tag{2}$$

   Top-$k$ documents based on $S_i$ are retrieved:

$$R_i = \text{TopK}(D_{\text{filtered}}, S_i). \tag{3}$$

The retrieved passages are compiled and passed to the language model to generate a coherent, evidence-grounded response. Weighting across fields allows the system to prioritize more informative content for the inferred intent. We evaluate the benefit of this component against a uniform-weight variant in Tab. 4.

# 4 Scientific Insights and Findings

Leveraging our LLM-driven multidimensional profiling framework, we analyze more than 100K papers accepted by 20+ premier AI conferences from 2020 to 2025. This section presents a series of quantitative and semantic analyses across multiple dimensions of the research ecosystem, including topic evolution, compute and model scale, dataset dynamics, and institution-level research specialization. These results together reveal both the macro-level shifts and the fine-grained emerging directions shaping modern AI research.

## 4.1 Topic Growth and Transitions

To characterize the long-term evolution of AI research topics, we analyzed topic dynamics from 2020 to 2025 using embedding-based clustering followed by LLM-assisted topic naming. We construct a four-quadrant topic lifecycle model, where the horizontal axis represents the Compound Annual Growth Rate (CAGR) of papers over the past two years, reflecting topic popularity trends:

$$\text{CAGR}_t = \left( \frac{N_t}{N_{t-2}} \right)^{\frac{1}{2}} - 1, \quad t = 2025. \tag{4}$$

The vertical axis represents the normalized mean publication year $\overline{Y}$ of papers, capturing research novelty:

$$\overline{Y}_i = \frac{1}{N_i} \sum_{p \in \text{topic } i} Y_p. \tag{5}$$

The bubble size indicates the total number of papers in the past two years $N_i$, and the color encodes a weighted impact metric combining citations $C_i$ and topic count $T_i$:

$$\text{W}_i = \frac{\alpha C_i + \beta T_i}{\max_j (\alpha C_j + \beta T_j)}, \quad \alpha + \beta = 1. \tag{6}$$

$\alpha$ and $\beta$ control the relative weights of citations and paper count, with $\alpha = 0.6$ in our experiments.

By analyzing the distribution of various topics in the four quadrants, as shown in Fig. 4, we observe that AI research is accelerating its evolution from being driven by early "model scale" and "basic perception tasks" to a paradigm driven by "safety and controllability, multimodal cognition, and intelligent agent systems". To further verify the overall trend of the life cycle analysis, we counted the changes in the number of papers on various topics over the past five years, and also selected the Top-20 topics with the largest number of papers for time series analysis, as shown in the lower left in Fig. 4. The topic identifiers in the inset correspond to the topic names listed in Appendix C; for example, topic 19 corresponds to efficient vision transformers and token mixing, which explains its consistently high citation signal across multiple years.

The results show that the paper volume of LLM-related topics exhibits exponential growth, especially in reasoning, long-context modeling, and preference alignment, with each surpassing 200 papers per year by 2025. Multimodal integration tasks such as text-to-image editing, video diffusion, and vision-language QA maintain high activity but show slight stabilization after their 2023 to 2024 peaks. Meanwhile, classical areas including neural machine translation and GNN-based methods display relatively flat or declining trajectories, consistent with a maturing stage. Safety and robustness topics, though more recent, feature the steepest growth curves in both publication count and citation acceleration, marking the next frontier of attention.

Overall, from the dual perspectives of life cycle and quantity trends, the analysis reveals a clear paradigm transition in current research topics, from model architecture optimization toward the integration of perception, reasoning, and interaction capabilities. Future directions that deserve particular attention include:

- Enhancing long-context and reasoning capabilities of foundation models through efficient memory and adaptive attention mechanisms;

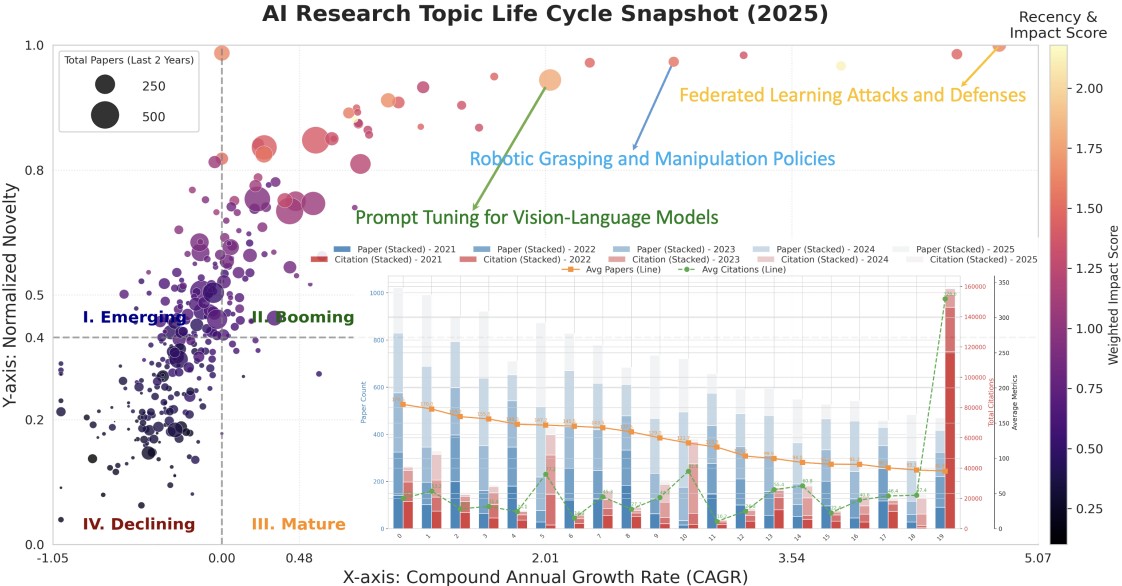

Figure 4: Four-quadrant topic lifecycle evolution (quadrants I-IV: Emerging, Booming, Mature, Declining). The inset (bottom left) shows the most prolific topic by paper count from 2020-2025.

- Strengthening safety, reliability, and alignment frameworks to ensure trustworthy human-AI collaboration;

- Advancing multimodal understanding and generation toward dynamic, interactive, and grounded cognition;

- Bridging efficiency and scalability with sustainable model deployment, including low-rank adaptation, quantization, and sparse computation strategies.

These directions exhibit both high recent growth rates and strong Weighted Impact values, suggesting sustained momentum in the evolving research landscape.

## 4.2  Compute & Model Scale Analysis

With the rise of large models and the increase in complex tasks, compute usage and model scale have become important dimensions for measuring research trends. We analyze only papers that explicitly report GPU or training-resource information and normalize the reported resources into A100 Equivalent Hours, as shown in Fig. 5. Papers without explicit compute information are stored as empty/unknown and are not imputed. Therefore, this analysis should be interpreted as the trend of *reported* compute usage among papers that disclose compute information, rather than a direct estimate of compute demand for all papers in the corpus. The resulting curve suggests an upward trend in reported training resources, especially for generative models, multimodal models, and agent-oriented systems, but this trend may also reflect increasing compute-reporting rates in recent papers.

Our research found that topics requiring high computing power investment have the following characteristics.

- **Processing complex data structures or large-scale data:** Research such as multimodality (combining images, text, speech, etc.) and large-scale pre-training involves massive amounts of training data with complex interrelationships between data, requiring more computing resources for feature extraction and model fitting.

- **Enlargement and deepening of model architectures:** Represented by large language models (LLMs) and large generative models, the number of parameters often reaches tens of billions or hundreds of billions, which directly leads to an exponential growth in the demand for computing resources during the training and inference phases.

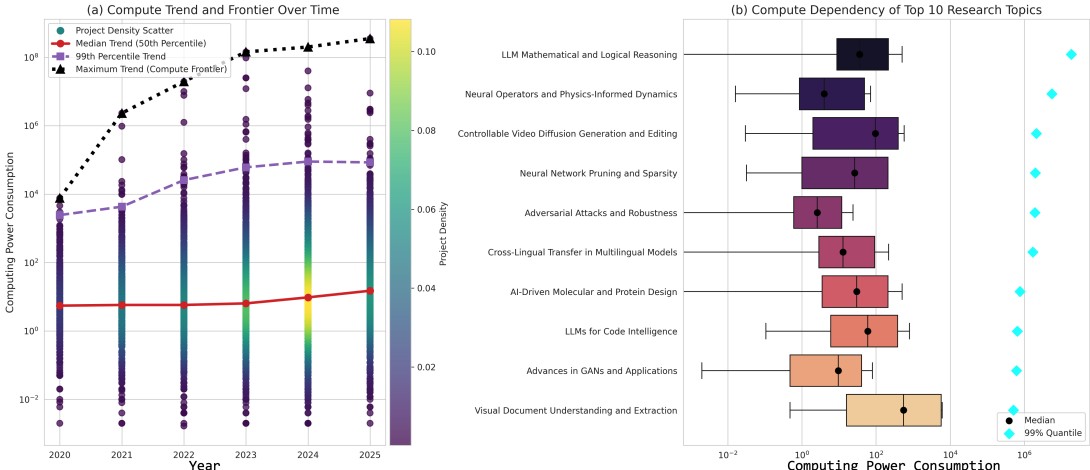

Figure 5: Compute Resource Analysis. Left: temporal evolution of reported compute usage among papers that disclose GPU or training-resource information; Right: boxplot of the top 10 topics with highest reported compute requirements.

- **Complex learning paradigms or training methods:** For example, research involving reinforcement learning (RL), adversarial learning (such as GANs), or research on intelligent agents that require a large number of interactions with simulated environments. These methods usually require multiple iterations, a large number of samples, or complex optimization processes, thereby greatly increasing computing power consumption.

Further analysis of the proportion of computing power and paper output across different topics reveals that the investment in computing resources shows an uneven distribution across various research directions:

- **High-investment and high-output fields:** Concentrated in cutting-edge areas such as generative models and multimodality. Although they consume the majority of equivalent A100 hours, they also yield the highest number of papers and the most influential research results, forming a positive cycle where computing power drives innovation.

- **High-investment and medium-to-low-output fields:** Research in certain intelligent agent or specific application domains may require enormous computing power for experiments due to immature methods or the inherent difficulty of tasks (such as exploration, the complexity of environmental interactions, etc.). However, the short-term paper output is relatively mismatched with their computing power consumption, which may represent future potential directions or bottlenecks in technological breakthroughs.

- **Low-investment and high-efficiency fields:** Areas such as model compression, efficiency optimization, and interpretability focus on algorithms and efficiency rather than the absolute scale of models. These studies achieve a relatively high number of paper outputs with relatively low computing power investment, and are of great value for the promotion and practical application of existing models.

Overall, among papers that disclose compute information, reported compute usage continues to rise, mainly driven by large-scale models and complex tasks. Generative models, multimodality, and intelligent agents are the fields with the most intensive reported compute investment, indicating likely future research hotspots. At the same time, this conclusion is conditional on compute reporting practices: because earlier papers often omit hardware or training-time details, the temporal pattern may conflate actual compute growth with increased reporting. We therefore treat the compute analysis as indicative evidence and discuss this limitation further in Sec. 6.

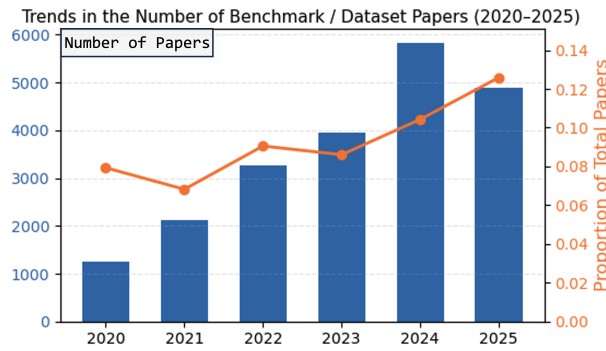

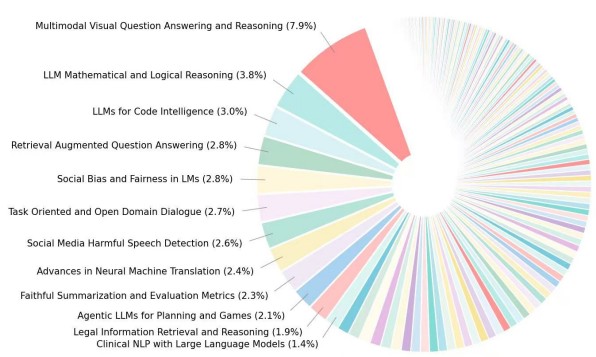

(a) Dataset and Benchmark Trends (2020-2025). The bar chart shows annual counts of dataset/benchmark-related papers, and the line chart shows their proportion of total publications.

(b) Topic distribution in Dataset/Benchmark papers. The most frequent topics are labeled.

Figure 6: The Analysis from the Dataset Landscape.

## 4.3 Dataset Landscape

Benchmark papers lay a crucial foundation for the standardization and comparability of technological progress by providing unified evaluation criteria and datasets. Statistics from the sample set show that the proportion of Benchmark papers is on a significant upward trend, with the growth rate accelerating particularly after 2022, as shown in Fig. 6a. Their proportion rose rapidly from 10.07% in 2022 to 18.16% in 2025 (partial data), approaching one-fifth of the total number of papers. This explosive growth is mainly due to the rise of general artificial intelligence technologies represented by LLMs, as well as the academic community's emphasis on the standardized evaluation of non-performance indicators such as model fairness, ethics, and robustness.

Further in-depth analysis of the themes of Benchmark articles reveals the evaluation challenges faced by current research hotspots and cutting-edge technologies. As shown in the statistical chart of theme count and proportion (Fig. 6b), research focuses are highly concentrated on LLMs, multimodality, and complex reasoning tasks. The top ten themes are almost entirely centered around the latest advancements and applications of LLMs, fully reflecting the evaluation-driven characteristics of contemporary scientific research.

Against the backdrop of this surge in benchmarks, the frequency of use of various datasets in scientific research papers has also shown rapid growth. This reflects the need for deeper evaluation in the era of LLMs and signifies that the paradigm for using datasets is no longer limited to single tasks but is evolving toward cross-domain and complex capability assessment, as shown in Fig. 7a

- **Plateauing classic visual benchmarks and possible cross-modal reuse.** Although ImageNet has long been the most widely used classic visual dataset, its usage appears to plateau after its 2022 peak rather than sharply decline. This pattern may indicate a maturing role for traditional visual benchmarks. In contrast, datasets such as COCO, which were originally designed for visual tasks such as object detection and instance segmentation, are increasingly reused in multimodal capability evaluation. We treat this ImageNet–COCO contrast as a suggestive observation: large-model research may be promoting cross-task and multi-purpose use of classic datasets, but further task-level validation is needed before drawing a definitive conclusion.

- **Explosive growth of high-level reasoning datasets.** As LLMs mature in basic language capabilities, research focus has shifted to complex cognitive and reasoning tasks. The number of papers using the MATH dataset, designed for evaluating mathematical reasoning and arithmetic abilities, as well as MMLU and GSM8K for high-level logical reasoning, has risen sharply since 2023. This trend strongly suggests that researchers are increasingly relying on these datasets to evaluate the performance of large models in complex reasoning tasks such as logic, common sense, and the integration of professional knowledge.

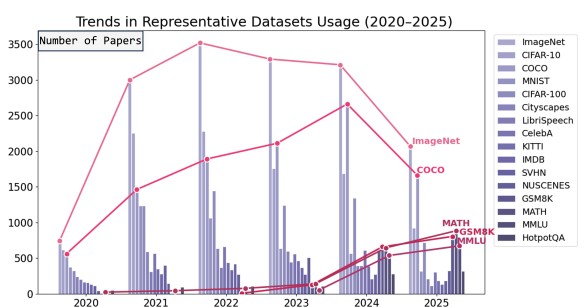

(a) Trends in Representative Dataset Usage (2020-2025). The grouped bar chart shows annual paper counts per dataset, with overlaid lines highlighting trends for MMLU, GSM8K, ImageNet, COCO, and MATH.

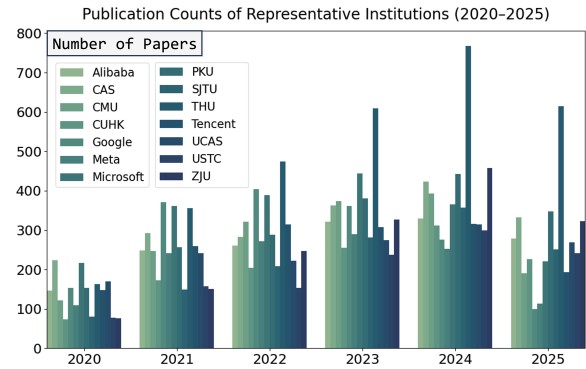

(b) Publication Counts of Leading Institutions (2020-2025). Shows the number of papers from representative academic and industrial institutions across 22 conferences.

Figure 7: Trends in Publication Counts for Dataset Usage and Leading Institutions from 2020 to 2025.

### 4.4 Institution-Level Research Patterns

Core institutions of China and the United States mainly maintained a dominant position across the period, as shown in Fig. 7b. Tsinghua University has long been at the top in terms of the number of publications, demonstrating the stability of its research output and its ability to continuously accumulate. However, there were annual competitive dynamic changes in the rankings between industrial giants and universities. In terms of the cooperation network, as shown in Fig. 8, domestic universities have active collaborations with industrial giants such as Alibaba, Tencent, and Huawei, showing obvious regional characteristics. At the same time, Chinese and American institutions still dominate the cooperation network, forming a "core-multipoint" cooperation pattern. This shift in research paradigms, driven by LLMs and Benchmark, is also profoundly reflected in the research strategies of different institutions. In the AI field, the academic community and the industrial sector have their own focuses and develop in a complementary manner. This is not simply a difference in interests, but rather reflects the strategic differentiation in the ecological niche of scientific research:

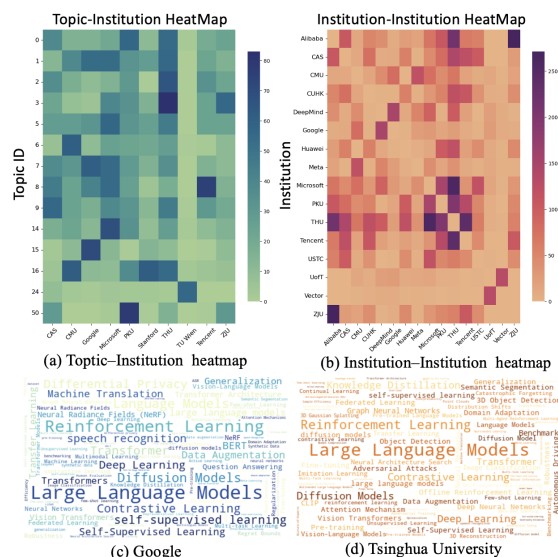

Figure 8: (a)–(b) Top topic-institution and institution-institution heatmaps; (c)–(d) Research topic word clouds for Google and Tsinghua University.

- **Academic institutions typically focus on basic research and cutting-edge algorithm exploration**, with their strategic priorities lying in efficiency improvement, robustness assurance, and continuous investigation of theoretical mechanisms, aiming to advance the democratization of AI technology. For instance, Tsinghua University emphasizes directions such as knowledge distillation, graph neural networks, adversarial training, domain adaptation, and model generalization, while Carnegie Mellon University demonstrates strong performance in areas like robotic grasping and manipulation strategies and causal discovery.

- **Industrial institutions lean more toward application-oriented research and theoretical studies related to technology deployment.** Leveraging their advantages in data and computing resources, they strive to address issues such as system efficiency and real-time performance in large-scale model deployment (Scaling-up) and commercial applications, establishing a complete AI

Table 3: Numeric expert scores corresponding to Fig. 9. Scores are reported on a 5-point scale.

| Method | Acc. | Cov. | Nov. | Read. | Use. | Mean |
|---|---|---|---|---|---|---|
| ChatGPT-5 Direct | 3.80 | 4.00 | 3.00 | **4.60** | 3.20 | 3.92 |
| SurveyX | 3.90 | **4.30** | 3.50 | 4.20 | 3.80 | 3.94 |
| *ResearchDB* | **4.50** | 3.90 | **4.20** | 4.00 | **4.40** | **4.20** |

ecosystem and technological moat. For example, Microsoft maintains leadership in LLMs inference, code intelligence, mathematical reasoning, and retrieval-augmented generation, while Google focuses more on application and system-level topics, including speech recognition, machine translation, multi-task learning, and federated learning.

# 5 Retrieval and Analysis

To evaluate the effectiveness of our retrieval-augmented LLM framework for semantic understanding and literature summarization, we selected the Top-20 topics based on paper volume and diversity across 22 AI conferences from 2020 to 2025. For each topic, papers were processed using two methods: (i) direct generation with ChatGPT-5, and (ii) retrieval-augmented generation leveraging our *ResearchDB*. ChatGPT-5 is used as a representative baseline, while the retrieval-augmented approach can be applied to other LLMs and assessed in future work. Evaluation was performed by five doctoral students along five dimensions:

- **Accuracy**: Consistency of the generated content with the original papers;

- **Coverage**: Inclusion of core content and major sub-directions within the topic;

- **Novelty**: Ability to capture recently emerging directions or methods;

- **Readability**: Clarity and logical coherence of the text;

- **Usefulness**: Facilitation of rapid understanding and insight into potential research trends.

Each criterion was scored on a 5-point Likert scale independently by all annotators, and the Fleiss' $\kappa$ value is 0.71, indicating substantial agreement among the five annotators.

The evaluation results in Fig. 9 and Tab. 3 reveal that our proposed framework outperforms the baseline methods across several critical dimensions. Compared to direct generation by ChatGPT-5, our approach achieves significantly higher Accuracy and Usefulness. This improvement is primarily attributed to our intent-driven hierarchical retrieval, which grounds the model's responses in structured evidence extracted from specific content fields. While SurveyX Liang et al. (2025) focuses on general automation, our framework excels in capturing technical specifics through weighted multi-field search. This evidence-grounded approach prioritizes high-fidelity sections (e.g., methods, datasets), improving reliability and traceability over unconstrained LLM generation, which is essential for high-stakes research analysis.

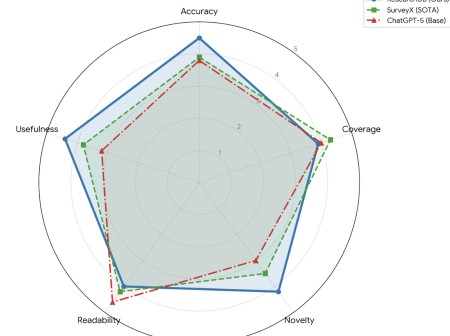

Figure 9: Performance comparison of *ResearchDB* against the SurveyX method and ChatGPT-5 across five evaluation dimensions, based on expert scoring.

## 5.1 Ablation Study

To evaluate the contribution of each core component in our hierarchical retrieval pipeline, we conduct an ablation study by comparing the full ResearchDB framework

Table 4: Ablation results on retrieval performance based on expert scores.

| Variant | Acc. | Cov. | Nov. | Read. | Use. | Mean |
|---|---|---|---|---|---|---|
| w/o MF | 4.12 | 3.85 | 4.00 | 4.10 | 3.90 | 4.01 |
| w/o IFW | 3.86 | 3.82 | 3.70 | 4.05 | 3.75 | 3.84 |
| SF | 3.70 | 3.10 | 3.20 | **4.20** | 3.25 | 3.49 |
| **ResearchDB** | **4.50** | **3.90** | **4.20** | 4.00 | **4.40** | **4.20** |

against three degraded variants: a configuration without metadata filtering (w/o MF), a version without intent-guided query-specific field weighting (w/o IFW) using a uniform weight distribution, and a single-field baseline (SF) that restricts semantic search to abstracts only. As shown in Tab. 4, the full framework achieves the highest mean score and leads in accuracy, coverage, novelty, and usefulness, while it is not the best on readability. The higher readability of SF and w/o MF likely reflects the fact that narrower or less constrained retrieval can produce simpler prose, whereas ResearchDB prioritizes evidence grounding and technical specificity. Specifically, replacing query-specific field weights with uniform weights prevents the system from effectively prioritizing task-relevant fields. Excluding metadata filtering reduces precision because the system may include papers from irrelevant venues or timeframes that share similar semantic embeddings. Finally, the weaker mean score of the single-field baseline confirms that relying solely on abstracts is insufficient for deep knowledge profiling, as essential information about datasets, training setup, and limitations is often omitted or overly generalized in paper summaries.

## 6 Conclusion

We present a multidimensional knowledge profiling framework that integrates topic clustering, LLM-based semantic parsing, and hierarchical retrieval to analyze over 100K AI publications. By capturing topic dynamics, dataset trends, and institutional patterns, the framework enables evidence-driven discovery of methodological shifts and technical paradigms. Our approach provides both a macro-level landscape overview and fine-grained, topic-level insights, effectively supporting systematic trend analysis and research planning.

**Data and Code Availability.** To support inspection and reuse, we will open-source all code and data associated with ResearchDB, including the extraction and retrieval pipeline, prompts, schema, topic labels, analysis scripts, query examples, and lightweight demo scripts for querying the database.

**Limitations and Impact.** While our study offers a transparent, evidence-based perspective on research evolution, it has several scope and measurement limitations. First, the corpus is limited to major English-language AI venues, so the resulting landscape may under-represent non-English, regional, or lower-resourced research communities. Second, the framework depends on LLM-assisted parsing; although most semantic fields show strong validation scores, GPU information is less reliable because many papers only partially report hardware or training time. Third, compute trends should be interpreted as trends in reported compute usage among papers that disclose compute information, not as absolute compute demand across the entire corpus, because reporting practices themselves change over time. Finally, automated literature analysis may create popularity feedback loops if users treat frequently retrieved or fast-growing topics as inherently more important than smaller but scientifically valuable directions. Future work will improve domain-specific parsing accuracy, report field coverage alongside trend statistics, and extend the framework to a broader range of scientific venues.

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

## Appendix

### Overview

The appendix presents the following sections to strengthen the main manuscript:

— We provide more details in the Supplementary Materials for a better understanding of our work.

— **Sec.** A. *ResearchDB* Source .

— **Sec.** B. Implementation and Reproducibility.

— **Sec.** C. Topic Clustering Results.

— **Sec.** D. Extraction Accuracy Validation.

— **Sec.** E. Retrieval System Evaluation.

## A *ResearchDB* **Source**

This appendix details the source, scale, and temporal distribution of the underlying database, as well as the multidimensional data schema used to construct the *ResearchDB* knowledge graph.

**Database Source and Distribution.** The *ResearchDB* comprises over 100,000 papers from 22 top-tier conferences and journals published between 2020 and 2025. This broad selection ensures comprehensive coverage across key AI and Computer Science sub-domains, as shown in Tab. A1.

Table A1: The 22 major venues in the ResearchDB corpus, categorized by domain, including their abbreviations and total paper count (2020-2025).

| Domain | Venue Name | Abbr. | Total Papers |
|---|---|---|---|
| Computer Vision | Conference on Computer Vision and Pattern Recognition | CVPR | 11436 |
| | International Conference on Computer Vision | ICCV | 3768 |
| | European Conference on Computer Vision | ECCV | 5389 |
| NLP & Speech | Annual Meeting of the Association for Computational Linguistics | ACL | 10595 |
| | Conference on Empirical Methods in NLP | EMNLP | 8703 |
| | North American Chapter of the ACL | NAACL | 3625 |
| | International Conf. on Comp. Linguistics (Modeling) | COLM | 299 |
| | International Speech Communication Association | INTERSPEECH | 5351 |
| Core ML & AI | Conference on Neural Information Processing Systems | NeurIPS | 12726 |
| | International Conference on Machine Learning | ICML | 10199 |
| | International Conference on Learning Representations | ICLR | 9512 |
| | Association for the Advancement of Artificial Intelligence | AAAI | 13575 |
| | International Joint Conference on Artificial Intelligence | IJCAI | 4263 |
| | Conference on Uncertainty in Artificial Intelligence | UAI | 1245 |
| | Conference on Robot Learning | CoRL | 814 |
| | International Workshop on Spoken Language Translation | IWSLT | 187 |
| | Conference on Learning Theory | COLT | 755 |
| Systems & Security | Machine Learning and Systems | MLSYS | 220 |
| | USENIX Symp. on Networked Systems Design and Implementation | OSDI | 258 |
| | Network and Distributed System Security Symposium | NDSS | 409 |
| | USENIX Conf. on File and Storage Technologies | USENIX-Fast | 165 |
| | USENIX Security Symposium | USENIX-Sec | 988 |
| | | **Overall Total** | $\approx 104482$ |

**Schema Usage and Worked Example.** The schema in Tab. A2 is used consistently across extraction, storage, and retrieval. During extraction, each paper is converted into one JSON

Table A2: Complete field-level schema for ResearchDB. The main text summarizes these fields by group in Tab. 2.

| Field Name | Field Group | Origin | Purpose/Description |
|---|---|---|---|
| paper_name, authors | Info | Metadata | Official title and author list. |
| conference, year | Info | Metadata | Publication venue and year. |
| institution | Info | Metadata | Author affiliations normalized for institution-level statistics. |
| topic_name, topic_ID | Info | Metadata | Hierarchical topic label assigned via clustering and LLM-assisted topic naming. |
| abstract_ori | Summary | Extracted | Original abstract extracted verbatim from the paper. |
| abstract_summary | Summary | LLM-assisted | Compressed summary of the original abstract. |
| keywords | Summary | Extracted | Technical terms extracted directly from the paper text. |
| keywords_description | Summary | LLM-assisted | Explanation or description for each extracted keyword. |
| problem_statement | Summary | LLM-assisted | Main problem or research question addressed by the paper. |
| contributions | Summary | LLM-assisted | Primary claimed contributions of the paper. |
| methods | Technical | LLM-assisted | Core algorithmic approach or methodology. |
| architecture | Technical | LLM-assisted | Model architecture or system design, when explicitly reported. |
| loss_function | Technical | LLM-assisted | Training objective or loss function, when reported. |
| datasets | Technical | LLM-assisted | Datasets used for experimental validation and benchmark analysis. |
| metrics | Technical | LLM-assisted | Evaluation metrics reported in the experiments. |
| datasets_metrics_mapping | Technical | LLM-assisted | Mapping between datasets and metrics for benchmark analysis. |
| experiments, results_summary | Analysis | LLM-assisted | Experimental setup and main empirical findings. |
| limitations, future_work | Analysis | LLM-assisted | Stated limitations and future directions. |
| novelty_type, field_positioning | Analysis | LLM-assisted | Coarse innovation type and field-level role of the paper. |
| trend_insight | Analysis | LLM-assisted | Short trend-level interpretation used for aggregate analysis. |
| training_setup | System | LLM-assisted | Optimization and resource-related training configuration. |
| gpu_info | System | LLM-assisted | Explicitly reported GPU or compute information. Missing values are stored as empty/unknown and are not inferred. |

record containing the fields in the authoritative schema. During storage, metadata fields such as `conference`, `year`, `authors`, `institution`, and `topic_ID` are indexed for filtering, while semantic fields such as `abstract_summary`, `keywords`, `keywords_description`, `problem_statement`, `contributions`, `methods`, `architecture`, `training_setup`, `datasets`, `metrics`, `gpu_info`, `experiments`, `results_summary`, `limitations`, `future_work`, and `field_positioning` can be embedded or routed for multi-field retrieval when relevant to the query.

For example, given the query "How did datasets for multimodal reasoning change in CVPR and NeurIPS from 2021 to 2025?", the system first extracts metadata filters `conference=[CVPR, NeurIPS]` and `year=[2021,...,2025]`. It then identifies the intent as a dataset/benchmark trend query and generates a weighted vector-search plan, assigning higher weight to `datasets` and `metrics`, with smaller weights on `keywords`, `abstract_summary`, and `methods`. The retrieved records are therefore constrained by venue and year, ranked by field-weighted semantic similarity, and summarized using the retrieved evidence rather than unconstrained model memory.

## B  Implementation and Reproducibility.

This section details the prompts and configurations used by the main implementation components of our large-scale multidimensional knowledge profiling framework: LLM-assisted knowledge parsing, intent-guided retrieval planning, field-weighted semantic search, and multidimensional topic clustering.

**LLM Knowledge Parsing Prompts.** We utilized the Deepseek-R1-32B model for semantic parsing of full paper texts to populate the *ResearchDB* schema (as documented in Sec. A). The parsing fidelity hinges on a carefully designed JSON-constrained prompt template. Below is the primary prompt template used for extracting and summarizing non-metadata fields. Bibliographic metadata such as title, authors, venue, and year is populated from venue records when available.

```
You are a scientific paper analysis assistant.

Your task is to read the input paper content (in Markdown format), and output a structured summary in JSON format
    according to the schema below.
```

```
Output JSON schema:
{
  "abstract_ori": "",
  "abstract_summary": "",
  "keywords": [],
  "keywords_description": {},
  "methods": "",
  "architecture": "",
  "loss_function": "",
  "training_setup": "",
  "gpu_info": "",
  "datasets": [],
  "metrics": [],
  "datasets_metrics_mapping": {},
  "problem_statement": "",
  "contributions": [],
  "novelty_type": "",
  "experiments": "",
  "results_summary": "",
  "limitations": [],
  "future_work": [],
  "trend_insight": "",
  "field_positioning": "",
  "institution": []
}

Instructions:
1. **Keywords**: Extract 3-10 keywords directly from the paper text. Use `keywords_description` to store a short LLM-
     generated explanation for each keyword.
2. **abstract_ori / abstract_summary**: Store the original abstract text in `abstract_ori` when it is available in
     the input. Provide a compressed, non-verbatim summary in `abstract_summary`.
3. **Methods / Architecture / Loss / Training**: Summarize core method(s), model architecture, loss function, and
     training setup.
4. **Datasets / Metrics / Mapping**: List datasets, corresponding metrics, and mapping between datasets and metrics.
     Use empty lists/dictionaries if missing.
5. **Problem Statement**: Summarize the research problem in 1-2 sentences.
6. **Contributions**: Summarize the core contributions.
7. **Novelty Type**: Infer innovation type:
   - Algorithm / Model
   - Theory / Analysis
   - Benchmark / Dataset
   - Application / System
   - Methodological Improvement
8. **Experiments / Results Summary**: Summarize experiments and main results. Include ablation studies if present.
9. **Limitations / Future Work**: Summarize limitations and future research directions.
10. **Trend Insight**: Provide trend level insights based on the paper.
11. **Field Positioning**: Infer the paper's position in its research field:
    - Foundational Work
    - Methodological Innovation
    - Benchmark / Dataset Contribution
    - Application Validation
    - Trend Extension
12. **Institution**: Extract all author affiliations from the paper when available.
13. **Gpu_info**: Extract explicitly reported GPU or compute resources in the format <total_gpu>*<GPU_MODEL>*<
     training_time>. If unavailable, return an empty string.
```

**Query Intent Recognition Prompt.** Following the original two-step retrieval design, the first prompt converts a natural-language query into metadata constraints, query terms, and candidate ResearchDB fields. It does not assign final weights. This separation keeps intent recognition aligned with the full schema in Tab. A2 while leaving query-specific weighting to the second prompt.

```
You are a comprehensive query analysis and search planning assistant for a scientific literature knowledge base.

Your primary task is to parse the user's natural language query into metadata filters, query terms, and candidate
     ResearchDB fields for later vector search. Do not recreate the full paper-level schema; choose only the fields
     relevant to the query intent.

Available searchable fields:
- abstract_summary
- keywords
- keywords_description
- problem_statement
- contributions
- methods
- architecture
- training_setup
- loss_function
- datasets
- metrics
- datasets_metrics_mapping
- gpu_info
- experiments
- results_summary
- limitations
```

```
- future_work
- novelty_type
- field_positioning
- trend_insight
- topic_name

Output Schema:
{
  "intent_type": "",
  "metadata_filters": {
    "conference": [],
    "year": [],
    "paper_name": [],
    "authors": [],
    "institution": [],
    "topic_name": [],
    "topic_ID": []
  },
  "query_terms": {
    "keywords": []
  },
  "candidate_search_fields": [
    {
      "field": "",
      "query_text": "",
      "reason": ""
    }
  ]
}

Rules:
1. **Metadata Filters**: Extract only explicitly mentioned metadata constraints, including conference, year,
     paper_name, authors, institution, topic_name, and topic_ID. Leave fields empty if they are not specified.
2. **Query Terms**: Extract technical terms explicitly mentioned in the query into 'keywords'. Field-specific search
     text should be written directly in each 'candidate_search_fields.query_text'.
3. **Candidate Search Fields**: Select relevant fields from the available searchable fields and write field-specific '
     query_text' for each selected field.
4. **No Hallucinated Constraints**: Do not invent metadata filters or field-specific query terms that are not implied
      by the user query.
5. **No Final Weights**: Do not assign weights in this step.
```

**Rule-Prior-Initialized Field-Weighting Prompt.** The second retrieval prompt refines the original field-weighting step by making the constraints explicit. Given the intent, metadata filters, query terms, and candidate fields from the first prompt, it produces a weighted vector-search plan. This step is neither a learned weighting model nor an unconstrained weight generator. Instead, it initializes field weights from intuitive cold-start priors summarized from typical query patterns and pilot retrieval experiments, then applies them only to the candidate fields selected by the intent-recognition step. Thus, the retrieval plan remains query-adaptive because different queries activate different fields, while the weighting behavior remains transparent and reproducible. The final selected fields must come from the candidate set, each field must be justified by the query intent, and the weights must sum to 1.0.

```
You are a rule-prior-initialized field-weighting assistant for a structured scientific literature database.

Given the user's query and the candidate_search_fields from the intent-recognition step, produce a weighted vector-
     search plan. Use the cold-start priors below as initial guidance, but only over the candidate fields selected
     for the current query.

Input JSON:
{
  "intent_type": "",
  "metadata_filters": {},
  "query_terms": {},
  "candidate_search_fields": [
    {
      "field": "",
      "query_text": "",
      "reason": ""
    }
  ]
}

Output JSON:
{
  "intent_type": "",
  "metadata_filters": {},
  "query_terms": {},
  "vector_search_plan": [
    {
      "field": "",
      "query_text": "",
      "weight": 0.0,
```

```
      "reason": ""
    }
  ]
}
```

Rules:
1. **Use Candidate Fields**: Assign weights only to fields in `candidate_search_fields`. Do not add fields that were not selected by the intent-recognition step.
2. **Cold-start Weighting Priors**: Use the following priors as reference values for common query intents. They are not learned parameters and not fixed routing tables:
   - Benchmark, dataset, or evaluation query:
     datasets 0.35, metrics 0.25, keywords 0.15,
     abstract_summary 0.15, methods 0.10.
   - Method, model, architecture, or algorithm evolution query:
     methods 0.35, architecture 0.25, keywords 0.15,
     abstract_summary 0.15, metrics 0.10.
   - Compute, GPU, training cost, or scaling query:
     gpu_info 0.40, training_setup 0.25, methods 0.15,
     datasets 0.10, metrics 0.10.
   - Limitation, future direction, or research-gap query:
     limitations 0.35, future_work 0.20, problem_statement 0.20,
     methods 0.15, keywords 0.10.
   - Topic trend, lifecycle, or emerging-direction query:
     keywords 0.25, abstract_summary 0.25, methods 0.20,
     datasets 0.15, metrics 0.15.
   - Institution, venue, author, or year-specific query:
     apply metadata filtering first, then assign semantic weights based on the remaining technical intent.
3. **Partial Overlap**: If the candidate fields only partially match a prior, keep the selected fields, discard absent fields, and renormalize the remaining prior weights. If a selected field has no direct prior but is clearly relevant, assign it a modest weight and justify the assignment.
4. **Mixed Intent**: If the query has multiple intents, combine the relevant priors according to the dominant intent and the selected candidate fields.
5. **Weight Normalization**: Normalize all weights in `vector_search_plan` so that their sum equals 1.0.
6. **Inspectability**: Each selected field must include a short reason explaining why it receives that weight.
7. **No Schema Bias**: Do not assign high weight to a field only because it exists in the schema.
8. **Fallback**: If the query is too broad or underspecified to justify differentiated weights, assign uniform weights over the selected fields and mark `intent_type` as `general_semantic_search`.

In-context example:
User query:
"How did datasets for multimodal reasoning change in CVPR and NeurIPS from 2021 to 2025?"

Output:
```
{
  "intent_type": "dataset_trend_analysis",
  "metadata_filters": {
    "conference": ["CVPR", "NeurIPS"],
    "year": [2021, 2022, 2023, 2024, 2025],
    "paper_name": [],
    "authors": [],
    "institution": [],
    "topic_name": [],
    "topic_ID": []
  },
  "query_terms": {
    "keywords": ["multimodal reasoning", "datasets"]
  },
  "candidate_search_fields": [
    {
      "field": "datasets",
      "query_text": "datasets used for multimodal reasoning",
      "reason": "The query primarily asks about dataset usage and changes."
    },
    {
      "field": "metrics",
      "query_text": "evaluation metrics for multimodal reasoning datasets",
      "reason": "Metrics help identify how datasets are used for evaluation."
    },
    {
      "field": "keywords",
      "query_text": "multimodal reasoning",
      "reason": "Keywords identify papers in the target technical area."
    },
    {
      "field": "abstract_summary",
      "query_text": "multimodal reasoning dataset trend",
      "reason": "Abstract summaries provide broad semantic context."
    },
    {
      "field": "methods",
      "query_text": "methods using multimodal reasoning datasets",
      "reason": "Methods help distinguish dataset use across task settings."
    }
  ]
}
```

```
Weighted output:
{
  "intent_type": "dataset_trend_analysis",
  "metadata_filters": {
    "conference": ["CVPR", "NeurIPS"],
    "year": [2021, 2022, 2023, 2024, 2025],
    "paper_name": [],
    "authors": [],
    "institution": [],
    "topic_name": [],
    "topic_ID": []
  },
  "query_terms": {
    "keywords": ["multimodal reasoning", "datasets"]
  },
  "vector_search_plan": [
    {
      "field": "datasets",
      "query_text": "datasets used for multimodal reasoning",
      "weight": 0.35,
      "reason": "The query primarily asks about dataset usage and changes."
    },
    {
      "field": "metrics",
      "query_text": "evaluation metrics for multimodal reasoning datasets",
      "weight": 0.20,
      "reason": "Metrics help identify how datasets are used for evaluation."
    },
    {
      "field": "keywords",
      "query_text": "multimodal reasoning",
      "weight": 0.20,
      "reason": "Keywords identify papers in the target technical area."
    },
    {
      "field": "abstract_summary",
      "query_text": "multimodal reasoning dataset trend",
      "weight": 0.15,
      "reason": "Abstract summaries provide broad semantic context."
    },
    {
      "field": "methods",
      "query_text": "methods using multimodal reasoning datasets",
      "weight": 0.10,
      "reason": "Methods help distinguish dataset use across task settings."
    }
  ]
}
```

**ResearchDB Topic Clustering Configuration.** The multidimensional topic structure of *ResearchDB* was generated through a robust embedding-based pipeline. We first transformed the paper abstracts and key textual segments into semantic vectors using the all-MiniLM-L6-v2 model. These high dimensional embeddings were then processed by UMAP for noise reduction and projection into a lower dimensional space, specifically configured with $n\_components = 40$, $n\_neighbors = 120$, and $min\_dist = 0.08$ (with a fixed $random\_state = 42$ for reproducibility). UMAP utilized the cosine metric, which aligns with the semantic nature of the embeddings. Clustering was performed using HDBSCAN, which is optimized for identifying clusters of varying densities; the algorithm was set with a significantly smaller $min\_cluster\_size = 50$ and $min\_samples = 1$, employing the `eom` (Excess of Mass) cluster selection method. Topic representation was established using C-TF-IDF, configured to filter vocabulary using MAX_DF_PERCENT = 0.9 and a MIN_DF_ABS_FLOOR = 50, utilizing an n-gram range of (1, 2). The full hierarchical structure, derived from the HDBSCAN tree and refined by an external LLM (ChatGPT-5) for descriptive naming, resulted in the fine-grained multi-level taxonomy of over 300 topics presented in Sec. C.

# C    Topic Clustering Results.

This section presents the complete results of the fine-grained topic taxonomy for ResearchDB. By combining sentence-transformers embeddings, UMAP dimensionality reduction, and the HDBSCAN clustering algorithm, we successfully identified and named 324 independent, high-precision research topics from the corpus of over 100,000 papers. The HDBSCAN configuration ensures that each topic represents a sufficiently large yet highly homogenous set of papers, guaranteeing the fine granularity and specificity of the topics.

**Hierarchical Topic Tree Structure.** A key advantage of the HDBSCAN algorithm is its inherent ability to produce a Hierarchical Structure. We leverage this structure as an organizational framework, providing a non-strict view of clustering relationships above the 324 fine-grained topics. This tree structure is the basis for our hierarchical retrieval system, allowing users to navigate from macro-level concepts (near the root) down to specific fine-grained topics (at the leaves), as shown in Fig. A1.

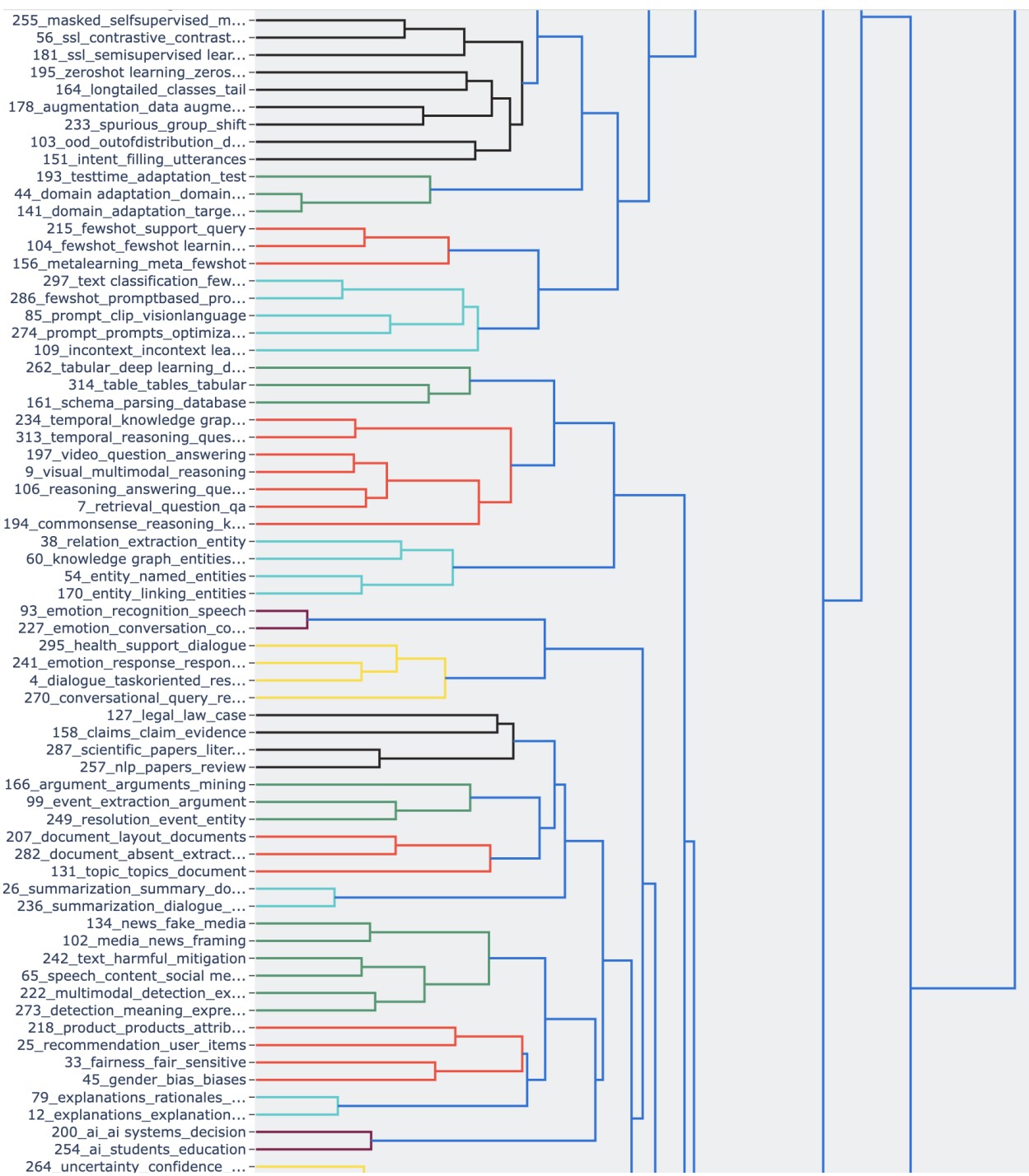

Figure A1: Hierarchical topic tree structure.

Table A3: Twenty Sample Fine-Grained Topics from *ResearchDB*, Illustrating the Taxonomy's Scope and Detail.

| ID | Topic Name | Summary |
|---|---|---|
| 0 | Graph Neural Networks and Representation Learning | Research advancing spectral and transformer-based GNNs, self-supervised and pretraining strategies, and scalable training to learn robust node and graph representations under heterophily, temporal dependencies, and distribution shifts. |
| 1 | Efficient Long-Context Attention and Memory | Research on extending and efficiently leveraging long contexts in transformers via improved positional encodings, attention approximations, adaptive computation, and memory augmentation, enabling length generalization and scalable training and inference. |
| 2 | Adversarial Attacks and Robustness | Research advances adversarial example generation and transferability across modalities and tasks while developing training, architectural, certification, and benchmarking methods to improve and assess deep models' robustness against digital, physical, and black-box attacks. |
| 3 | AI-Driven Molecular and Protein Design | Research leveraging structure- and sequence-informed machine learning, including graph and equivariant models, to predict properties, model interactions, and generate designs for molecules, proteins, and materials. |
| 4 | Task Oriented and Open Domain Dialogue | Research advances task-oriented and open-domain dialogue by combining end-to-end response generation with policy learning and state tracking, grounding on external knowledge and personas, handling complex goals, and developing robust evaluation. |
| 5 | LLM Mathematical and Logical Reasoning | Research advances methods and benchmarks to enhance and evaluate the mathematical and logical reasoning capabilities of large language models via chain-of-thought and structured prompting, neuro-symbolic and programming scaffolds, data/tool augmentation, self-correction, and reward modeling, and step-level and compositional evaluation. |
| 6 | Causal Discovery and Treatment Effects | Research develops methods to learn causal structure and estimate treatment and direct/indirect effects via interventions and counterfactuals, addressing hidden confounding, temporal dynamics, and identifiability in observational and experimental data. |
| 7 | Retrieval Augmented Question Answering | Research advances dense and generative retrieval and retrieval-augmented generation to enhance LLM question answering, addressing document identifiers, ranking and sampling, query rewriting and generation, knowledge selection, and rigorous evaluation for accurate and robust answers. |
| 8 | Advances in Neural Machine Translation | Research advances neural machine translation through improved architectures and decoding, multilingual and low-resource adaptation and transfer, robust evaluation and quality estimation, and human-in-the-loop and memory-assisted techniques. |
| 9 | Multimodal Visual Question Answering and Reasoning | Research advances vision-language models for multimodal VQA and reasoning, introducing datasets and benchmarks, alignment and retrieval strategies, and methods for compositional, chart, spatial, and knowledge-based tasks, and improving evaluation. |
| 10 | Text-to-Image Editing and Control | Research advances text-to-image diffusion with fine-grained controllability and personalization via improved architectures, guidance mechanisms, and language-vision integration, enabling precise editing, multi-concept synthesis, and preference-aligned generation. |
| 11 | Stochastic Bandits and Regret Bounds | Research develops algorithms and theoretical guarantees for stochastic, contextual, and linear bandits, including combinatorial, multi-agent, federated, and multi-objective variations, improving regret and sample complexity bounds under diverse feedback and constraints. |
| 12 | Explainable AI with Trees Concepts Counterfactuals | Research unifies decision tree modeling and optimization, prototype and concept-based self-explanation, Shapley/attribution and interaction metrics, and counterfactual and recourse frameworks to provide faithful, efficient, and user-relevant explainability. |
| 13 | Heterogeneity-Aware Personalized Federated Learning | This topic advances personalized FL by mitigating statistical and system heterogeneity through personalized models, heterogeneity-aware aggregation and alignment, and communication-efficient or asynchronous training, enhancing robustness, generalization, and scalability across clients and tasks. |
| 14 | LLMs for Code Intelligence | Research leverages LLMs with verification, search, and program analysis capabilities to generate, fix, translate, and retrieve code across languages, while benchmarking, evaluating robustness and bias, detecting hallucinations, and integrating developer feedback and tools. |
| 15 | Differential Privacy Algorithms and Utility Guarantees | Research advances differential privacy through novel mechanisms, optimization, and learning algorithms (e.g., adaptive clipping, shuffle amplification) to protect gradients and data in federated and centralized settings with formal guarantees and improved utility bounds. |
| 16 | Robotic Grasping and Manipulation Policies | Research develops learning-based policies for dexterous grasping and object manipulation, utilizing demonstrations, reinforcement learning, simulation, and generative models with multimodal sensing to achieve robust and generalizable control. |
| 17 | Temporal Action Recognition and Localization | Research on spatio-temporal modeling for recognizing and temporally localizing human actions in videos, emphasizing long-range dynamics, label-efficient learning (weak/semi/unsupervised), and robust representations against noise and backgrounds. |
| 18 | Agentic LLMs for Planning and Games | Research develops and evaluates LLM-based agents that plan, communicate, and act in games and interactive environments, integrating symbolic planning, multi-agent coordination, and experiential learning for robust decision-making. |
| 19 | Efficient Vision Transformers and Token Mixing | This topic advances efficient vision backbones by redesigning self-attention and token processing (via token pruning/merging, alternative mixers (linear/FFT), and state space models) to preserve global context while reducing computation and memory for classification, detection, and segmentation. |

**Topic Naming and Examples.** The naming of each fine-grained topic underwent strict quality control: C-TF-IDF was used to extract representative keywords, which were then analyzed and refined by ChatGPT-5 based on paper abstracts within the cluster. This process generated highly semantically clear Topic Names (e.g., 140_Approximation and Fair Clustering Algorithms). Every paper in *ResearchDB* is uniquely tagged with such a topic name. Tab. A4 below presents a sample of the most representative fine-grained topics. We also have selected topics with ids ranging from 1 to 20 and listed them as Tab. A3

Table A4: Sample list of fine-grained topics extracted from *ResearchDB*, illustrating the specificity and coverage of the taxonomy.

| Domain | ID | Topic Name | Representative Keywords |
|---|---|---|---|
| Optimization Theory | 140 | Approximation and Fair Clustering Algorithms | correlation clustering, approximation ratio, $\ell_p$ norm |
| Computer Vision | 088 | Efficient ViT Training and Pruning | vision transformer, token pruning, dynamic exit |
| Natural Language Processing | 215 | Controllable Text Generation via Latent Space | VAE, latent variable, attribute control, language model |
| Core ML | 042 | Causal Inference for Robustness and Generalization | do-calculus, invariant prediction, domain generalization |
| Systems & Security | 291 | Hardware Acceleration for Neural Network Inference | FPGA, ASIC, low-bit quantization, hardware optimization |

## D  Extraction Accuracy Validation.

This section validates the reliability of the structured fields in *ResearchDB*. Since different fields have different levels of ground-truth availability, we use two complementary checks: automatic verification for metadata fields and manual evaluation for semantic fields.

**Metadata Verification.** For fields with explicit ground truth, including paper title, authors, venue, and year, we compare the extracted records against the original venue metadata after basic normalization, such as lowercasing, whitespace normalization, and punctuation cleanup. The exact-match accuracy is reported in Tab. A5. This check mainly verifies whether the data alignment and extraction pipeline preserve the original bibliographic information.

Table A5: Automatic validation of metadata fields against original venue records.

| Field | Ground Truth Source | Metric | Accuracy |
|---|---|---|---|
| Paper Title | Venue metadata | Exact Match | 93.3% |
| Authors | Venue metadata | Exact Match | 95.2% |
| Conference | Venue metadata | Exact Match | 100.0% |
| Year | Venue metadata | Exact Match | 100.0% |

**Manual Evaluation of Semantic Fields.** For fields that do not have a single fixed answer, including keywords, methods, datasets, metrics, and GPU information, exact matching is not appropriate. We therefore manually evaluate 200 randomly sampled papers from different venues, years, and research areas. For each paper-field pair, annotators assign a score of 1 if the extracted content is correct and usable, 0.5 if it is partially correct or incomplete, and 0 if it is unsupported or assigned to the wrong field. The detailed scoring criteria are shown in Tab. A6.

We report the average score for each field and criterion in Tab. A7. This evaluation focuses on whether the extracted semantic fields are faithful and useful for downstream analysis, rather than whether the LLM produces the only possible wording of a summary.

Table A6: Manual scoring rubric for semantic-field extraction. Each criterion is scored as 1, 0.5, or 0 for each sampled paper-field pair.

| Criterion | Description |
|---|---|
| Factual Support | Whether the extracted item is directly stated in, or reasonably inferred from, the paper text. |
| Coverage | Whether the extraction includes the main items needed to represent the paper, rather than missing key methods, datasets, metrics, or resources. |
| Specificity | Whether the extracted content is concrete enough for downstream trend analysis, instead of being overly generic. |
| Field Consistency | Whether the item is placed in the correct schema field, such as datasets versus metrics or methods versus architecture. |
| Statistical Usability | Whether the extracted content can be used in aggregate statistics after normalization, especially for datasets, metrics, and GPU information. |

**Compute Field Coverage.** GPU information is treated differently from fields such as datasets or metrics because many papers do not report hardware, training time, or total accelerator count. The extraction pipeline therefore stores `gpu_info` as empty/unknown when compute resources are not explicitly reported in the paper, rather than inferring missing values. For this reason, compute-trend results in Fig. 5 should be interpreted as reported compute usage among papers with explicit compute disclosure, not as a corpus-wide estimate of absolute compute demand. We do not infer missing hardware information or add a separate reporting-rate-corrected growth analysis in this revision.

Table A7: Manual validation results on 200 randomly sampled papers. Scores are averaged over sampled paper-field pairs using the 0/0.5/1 rubric.

| Field | Factual Support | Coverage | Specificity | Field Consistency | Stat. Usability | Overall |
|---|---|---|---|---|---|---|
| Keywords | 0.90 | 0.84 | 0.87 | 0.93 | 0.86 | 0.88 |
| Methods | 0.86 | 0.80 | 0.83 | 0.91 | 0.82 | 0.84 |
| Datasets | 0.93 | 0.88 | 0.91 | 0.96 | 0.90 | 0.92 |
| Metrics | 0.89 | 0.82 | 0.86 | 0.94 | 0.87 | 0.88 |
| GPU Information | 0.76 | 0.62 | 0.78 | 0.88 | 0.68 | 0.74 |

**Qualitative Extraction Examples.** The successful retrieval and evidence-grounded synthesis example in Fig. A2 illustrates a typical case where the structured fields provide useful support for downstream analysis. The retrieved evidence contains relevant papers and technical descriptions, allowing the final answer to attribute methods and organize the research landscape more concretely than direct generation. We also include two representative failure cases in Fig. A3. These cases focus on practical limitations observed during manual validation: method summaries can be overly generic, and dataset extraction may miss secondary or transfer-evaluation datasets when many experimental settings are reported.

# E    Retrieval System Evaluation.

This appendix details the validation of the *ResearchDB* retrieval system. Since our goal is literature analysis and trend synthesis rather than single-hop question answering, a standard QA benchmark does not fully capture the intended use case. We therefore evaluate whether each system can produce a useful, evidence-aware analysis for complex research queries.

**Evaluation Protocol.** We construct 2,000 literature-analysis queries covering topic evolution, benchmark usage, methodological changes, and compute/resource trends. For each query, we compare three systems: direct ChatGPT-5 generation, SurveyX, and our *ResearchDB*-supported method. ChatGPT-5 answers directly from its parametric knowledge. SurveyX is used as a strong automated-survey baseline. *ResearchDB* first retrieves papers and structured fields from the database, then generates the final analysis conditioned on the retrieved evidence.

To avoid the bias and limited scale of small human-only evaluation, we use an LLM-as-judge protocol. For each query, we feed the judge model the query, the generated answer, and the retrieved or cited papers/statistics

used as evidence. The judge then scores the answer on a 5-point scale along five dimensions: **Accuracy**, whether the answer is consistent with the source papers; **Coverage**, whether it covers the main sub-directions required by the query; **Novelty**, whether it captures recent or emerging directions; **Readability**, whether the answer is coherent and easy to follow; and **Usefulness**, whether it helps readers understand the research landscape. The averaged results are shown in Tab. A8.

Table A8: Retrieval and analysis evaluation on 2,000 literature-analysis queries. Scores are averaged using an LLM-as-judge protocol on a 5-point scale.

| Method | Acc. | Cov. | Nov. | Read. | Use. | Mean |
|---|---|---|---|---|---|---|
| ChatGPT-5 Direct | 3.42 | 3.68 | 3.21 | **4.55** | 3.36 | 3.65 |
| SurveyX | 3.86 | **4.18** | 3.63 | 4.21 | 3.92 | 3.96 |
| *ResearchDB* | **4.27** | 4.05 | **4.11** | 4.08 | **4.31** | **4.17** |

**Evaluation Results.** Overall, *ResearchDB* achieves the best scores in accuracy, novelty, and usefulness. This suggests that structured retrieval improves the factual grounding and practical value of generated literature analysis. SurveyX obtains the highest coverage score, reflecting its strength in broad survey-style organization. ChatGPT-5 direct generation remains strongest in readability, but it is weaker in accuracy, novelty, and usefulness because it lacks explicit access to recent retrieved papers and structured evidence. To illustrate this advantage, we conducted a qualitative assessment contrasting the *ResearchDB* output with a ChatGPT-5 baseline on a complex query: "How to achieve real-time, high-fidelity speech-to-gesture generation using decoupled diffusion models." The *ResearchDB*-supported answer (akin to an Academic Review) showed clear superiority in Fig. A2:

1. **Enhanced Academic Rigor and Traceability:** *ResearchDB* successfully retrieved seminal papers, allowing the final answer to cite up to nine top-tier conference papers (e.g., [1], [3], [19]), accurately attributing key methods like Retrieval Augmentation.

2. **Clearer Theoretical Framing:** The retrieved content enabled high-level theoretical decoupling of the problem (e.g., separating gesture drivers into semantic, rhythmic, and stylistic factors), resulting in a logically coherent survey argument.

This qualitative evidence confirms that *ResearchDB*'s multidimensional matching enhances the output's Academic Robustness, ensuring the retrieved information is both relevant and authoritative for writing top-tier research reviews.

**Efficiency.** To evaluate the scalability of ResearchDB, we benchmarked the knowledge profiling pipeline on a cluster equipped with 8 NVIDIA A100 (80GB) GPUs.

- **Knowledge Extraction Throughput:** By leveraging a distributed parallel processing framework with vLLM for inference acceleration, the system achieved a steady-state throughput of approximately 870 full papers per hour. The entire corpus of 104,482 papers was processed in 120 hours (5 days) of wall-clock time, totaling 960 GPU hours.

- **Data Sharing and Accessibility:** To facilitate future research and ensure reproducibility, we will release the fully structured ResearchDB, containing multidimensional profiles (e.g., methods, datasets, and limitations) for all 104,482 papers. This pre-processed knowledge base allows researchers to perform large-scale trend analysis and complex semantic queries without incurring the high computational costs of initial parsing.

- **Query Response:** For the end-user, the parallelized intent recognition module decomposes complex queries into sub-tasks that are executed concurrently, maintaining an average end-to-end response time of ~1.2 seconds.

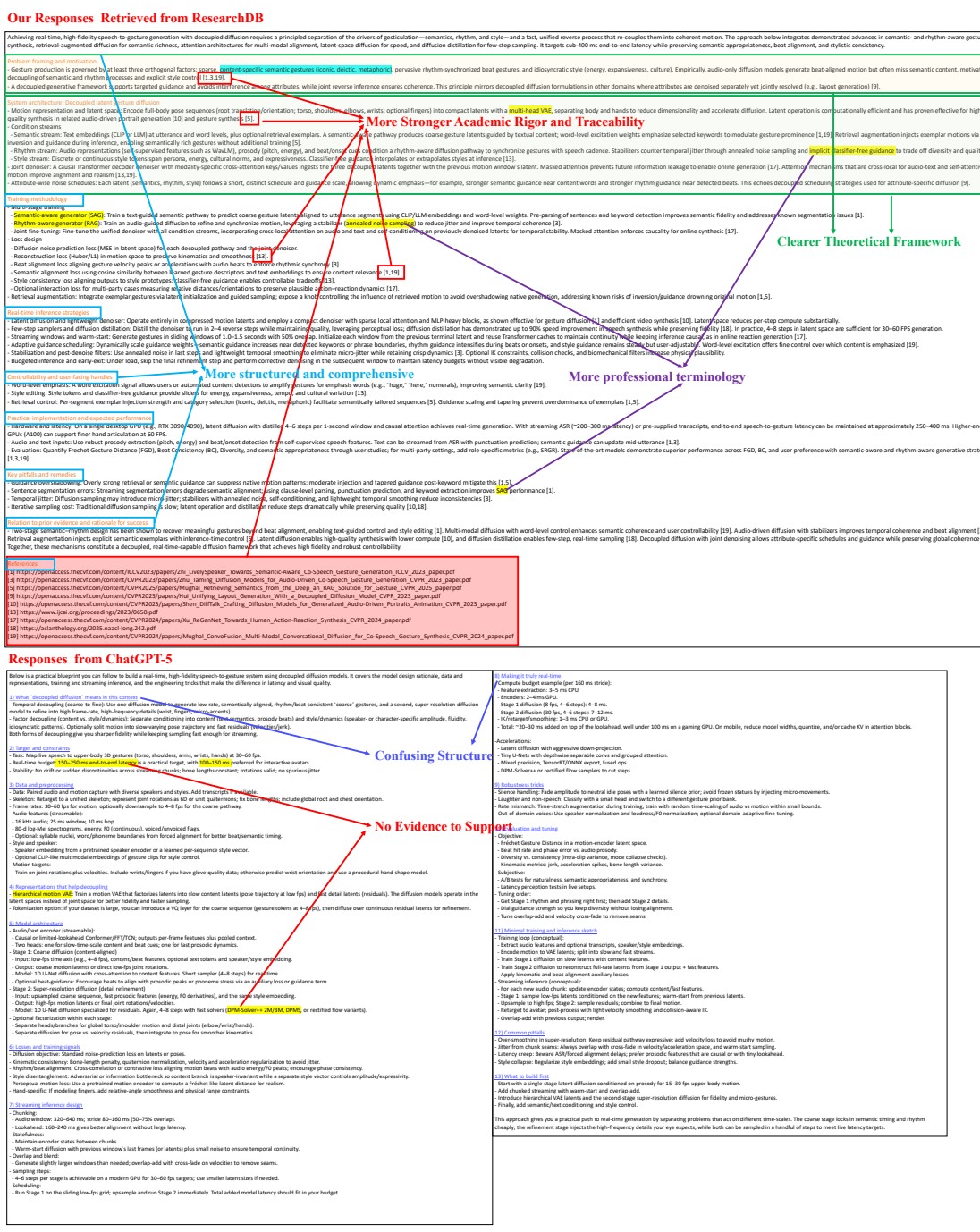

Figure A2: Qualitative comparison of answer quality between the *ResearchDB* system and the ChatGPT-5 baseline on the query: "How to achieve real-time, high-fidelity speech-to-gesture generation using decoupled diffusion models."

---

**Method extraction failure.**
**Paper evidence:** The paper proposes a decoupled generation framework with separate semantic, rhythmic, and style branches, together with a temporal consistency objective for improving motion coherence.
**Extracted field:** "The paper uses a diffusion-based generation model."
**Issue:** The extracted method is factually related but too generic. It misses the key design choices, including decoupled branches and temporal consistency modeling.

**Dataset extraction failure.**
**Paper evidence:** The experiments report results on HumanML3D and KIT-ML, and include an additional cross-dataset evaluation on a speech-gesture benchmark.
**Extracted field:** "HumanML3D."
**Issue:** The extraction captures only the main dataset and misses secondary or transfer-evaluation datasets, leading to incomplete dataset coverage for downstream statistics.

---

Figure A3: Representative failure cases in semantic-field extraction. The examples show generic method extraction and incomplete dataset coverage, two common practical limitations observed in manual validation.

