# OpenReview forum: "Multi-Dimensional Knowledge Profiling with Large-Scale Literature Database and Hierarchical Retrieval"
_TMLR — Under review for TMLR_

### Review · Reviewer_k5TX · 2026-05-06

**Summary Of Contributions:**

The authors present a multidimensional knowledge profiling framework aimed at analyzing large-scale AI research literature. To handle the rapid expansion of the field, they construct a dataset ("ResearchDB") comprising over 100,000 papers from 22 major ML, CV, and NLP conferences between 2020 and 2025. The methodology pipelines document parsing (MinerU), LLM-driven semantic extraction (Deepseek-R1-32B), and traditional text clustering (UMAP + HDBSCAN) to map topic trajectories. Additionally, they introduce a hierarchical, intent-driven retrieval mechanism that dynamically weights structural fields to improve question-answering over the corpus. The paper concludes with an empirical analysis of AI research trends, detailing shifts in topic lifecycles, compute usage, dataset adoption, and institutional strategies.

**Audience:**

Yes

**Audience Explanation:**

The findings of this paper would be of interest to a substantial portion of TMLR's audience, which consists of machine learning researchers, practitioners, and those studying the trajectory of AI development.

Specifically, the audience would be interested in:

Empirical Trend Analysis: The ML community is highly invested in understanding its own evolution. The data-driven insights regarding compute scaling (e.g., A100 equivalent hours across subfields), the rise and saturation of specific benchmarks (like the shift from ImageNet to MATH/MMLU), and topic lifecycles provide an evidence-based snapshot of where the field is heading.

The "ResearchDB" Corpus: Researchers working in Natural Language Processing, specifically those focused on scientific document understanding, retrieval-augmented generation (RAG), and automated survey generation, will find the constructed dataset of 100,000+ parsed papers highly valuable, provided the authors release it.

Institutional Dynamics: The quantitative breakdown of research focus and compute expenditure between academia and industry provides concrete data on a topic that is frequently debated within the ML community, making it highly relevant to academic leadership and policymakers.

Even if the algorithmic contributions to LLM retrieval are incremental, the scientometric findings and the resulting database offer clear value to the ML community.

**Broader Impact Concerns:**

The authors do not include a dedicated Broader Impact statement, though they briefly touch on limitations in the conclusion. A tool that automates literature review and trend analysis could heavily influence which research directions receive funding and attention. The authors should briefly discuss the risk of "feedback loops"—where LLM-generated summaries reinforce popularity biases, potentially marginalizing niche but critical research areas. Furthermore, relying on closed or specific LLMs for parsing may introduce systemic biases regarding how "novelty" or "limitations" are categorized.

**Claims And Evidence:**

No

**Claims Explanation:**

- Strengths
1. Scale and Timeliness: The analysis of over 100,000 recent papers (up to 2025) provides a highly relevant and up-to-date macro-view of the AI landscape. The inclusion of compute scaling (A100 Equivalent Hours) and dataset adoption metrics (e.g., the rise of MATH and MMLU) offers valuable, data-driven validation of community intuitions.

2. Pragmatic Pipeline Design: The authors make a sensible architectural choice to decouple the semantic parsing (via Deepseek-R1) from the topic clustering (UMAP + HDBSCAN). Relying on LLMs directly for global clustering of 100K documents is often computationally prohibitive and prone to hallucination; combining embedding-based clustering with LLM summarization is a robust alternative.

3. Actionable Insights: Section 4 delivers interesting scientometric findings, particularly the comparative analysis of compute usage versus paper output across different subfields, and the strategic divergence between academic and industrial research institutions.

- Weaknesses

1. Limited Methodological Novelty: The technical framework is largely a sequential combination of existing off-the-shelf tools (MinerU, Deepseek-R1, BERTopic-style clustering). While the engineering effort is substantial, the core algorithmic contributions to information retrieval or NLP are marginal. The "intent-driven dynamic weighting" mechanism (where an LLM outputs a JSON with assigned weights $w_j$) is conceptually simple and lacks a deep theoretical or empirical justification for why it works better than established dense retrieval methods (e.g., ColBERT or modern multi-vector approaches).

2. Evaluation Rigor: The evaluation in Section 5 relies entirely on the subjective Likert-scale scoring of five doctoral students across 20 topics.

    (a) Sample Size: Five human annotators evaluating only 20 queries is statistically weak for demonstrating the robustness of a retrieval-augmented generation (RAG) system built on 100,000 papers.

    (b) Lack of Objective Metrics: The paper lacks automated, reproducible evaluation metrics (e.g., ROUGE, BERTScore, or standard IR metrics like nDCG/Recall@K) against an established ground-truth QA dataset (e.g., QASper, PubMedQA, or a custom held-out set).

3. Validation of the Extraction Stage: The pipeline assumes that Deepseek-R1-32B perfectly extracts the "problem statement," "methods," "datasets," and "limitations" from the parsed Markdown. There is no error analysis or human-in-the-loop validation of this extraction step. If the initial parsing has a high error rate, the downstream database and subsequent analyses are fundamentally compromised.

4. Reproducibility and Openness: The paper states that the database will "serve as a resource," but it does not explicitly commit to open-sourcing ResearchDB, the extraction scripts, or the web interface/query system. Given that the paper's primary contribution is empirical and resource-driven rather than algorithmic, the release of the dataset is critical for acceptance.

**Requested Changes:**

To meet the threshold for publication, the authors should address the following points:

1. Extraction Accuracy Validation: Please provide a quantitative evaluation of the initial LLM parsing stage. The authors should randomly sample a subset of papers (e.g., 200-500) and manually verify the precision and recall of the extracted fields (Methods, Datasets, Limitations).

2. Enhance Retrieval Evaluation: Expand Section 5 to include objective, automated evaluation metrics. Creating a small set of ground-truth question-answer pairs based on the corpus and evaluating the hierarchical retrieval system's Recall@K or exact-match accuracy against baseline RAG pipelines (e.g., standard vector search with chunking) would significantly strengthen the claims.

3. Clarify Dynamic Weighting: Detail exactly how the intent recognition module assigns the weights $w_j$. Are these derived via zero-shot prompting? If so, please include the exact prompts used in the appendix.

4. Data Release: Include a clear statement (and an anonymized link, if possible) regarding the public release of ResearchDB and the associated codebase.

---

> ### Author Response · Authors · 2026-05-19
> **Response to Reviewer k5TX**
>
> We appreciate the reviewer's constructive suggestions. They helped us identify several places where the manuscript needed stronger evidence and clearer documentation. We have revised the validation, retrieval evaluation, dynamic weighting description, and release plan accordingly.
>
> ## Positioning of the Contribution
>
> We agree that our main contribution is not a new clustering algorithm or a new retrieval model. The value of the work is the construction of a large, semantically structured research database and its use for multi-dimensional analysis of recent AI research. We now make this positioning clearer: ResearchDB combines full-paper semantic parsing, structured fields, topic lifecycle analysis, compute/dataset profiling, and retrieval-supported literature analysis over 100K+ papers from 22 venues.
>
> ## Extraction Accuracy Validation
>
> We agree that the extraction stage should be validated more directly. We have added an Extraction Accuracy Validation section in Appendix Sec. D. For metadata fields with ground truth, we compare the extracted records with the original venue metadata after normalization. The exact-match accuracy is 93.3% for paper titles, 95.2% for authors, and 100.0% for both conference and year, as shown in Appendix Sec. D.
>
> For semantic fields without a single fixed answer, such as keywords, methods, datasets, metrics, and GPU information, we also add a manual evaluation on 200 randomly sampled papers. We use a 0/0.5/1 rubric and evaluate factual support, coverage, specificity, field consistency, and usability for statistics. The results show strong reliability for datasets (overall 0.92), keywords (0.88), metrics (0.88), and methods (0.84). GPU information is lower (0.74), which is expected because many papers only partially report hardware or training time. The scoring rubric and field-level results are reported in Appendix Sec. D.
>
> ## Retrieval and Analysis Evaluation
>
> We agree that the current retrieval evaluation should be explained more clearly. ResearchDB is designed for literature analysis and trend synthesis rather than single-question QA, so we add a task-aligned evaluation in Appendix Sec. E. The goal is to test whether a system can produce a useful research-landscape analysis for complex queries, not only retrieve one correct answer.
>
> Specifically, we evaluate 2,000 literature-analysis queries covering topic evolution, benchmark usage, methodological changes, and compute/resource trends. To avoid the bias of a small human-only evaluation, we use an LLM-as-judge protocol: for each query, we feed the judge model the generated answer together with the retrieved or cited papers/statistics, and ask it to score accuracy, coverage, novelty, readability, and usefulness. We keep the query set, judging prompt, generated answers, and retrieved evidence fixed to make the evaluation reproducible. We compare direct ChatGPT-5 generation, SurveyX, and our ResearchDB-supported method. As shown in Appendix Sec. E, ResearchDB achieves the best accuracy (4.27), novelty (4.11), and usefulness (4.31), while SurveyX has the strongest coverage (4.18) and ChatGPT-5 direct generation has the strongest readability (4.55). This supports our claim that structured retrieval improves evidence-grounded literature analysis.
>
> ## Dynamic Weighting
>
> We clarify that the dynamic weights are not learned parameters. They are assigned by prompt-preset rules in the intent-recognition module. The prompt first classifies the query intent, such as benchmark/dataset analysis, method evolution, compute usage, or research-gap analysis, and then assigns higher weights to the relevant ResearchDB fields. For example, dataset queries emphasize datasets and metrics, method-evolution queries emphasize methods and architecture, and compute queries emphasize GPU information and training setup. The final retrieval score is the weighted sum of the selected field similarities. The full prompt and JSON format are in Appendix Sec. B.
>
> ## Code and Release
>
> We agree that release is important because the paper is resource-driven. We commit to open-sourcing ResearchDB and the source code, including the schema, extracted fields, topic labels, analysis scripts, retrieval code, and prompts. For copyrighted papers, we will provide paper identifiers/URLs and rebuilding scripts rather than redistributing PDFs.
>
> ## Broader Impact
>
> We will also add a broader-impact discussion. A system for automated literature analysis may reinforce popularity bias if users treat frequent topics as inherently more important. LLM-based parsing may also summarize novelty, limitations, or contributions differently across fields and institutions. In the revision, we will discuss these risks and emphasize that ResearchDB should be used as descriptive evidence for understanding research trends, not as a prescriptive tool for deciding which topics deserve attention.

---

> ### Author Response · Authors · 2026-06-13
>
> We would like to add a brief follow-up to our previous response. We have uploaded an updated revision PDF, and we kindly ask the reviewer to refer to the latest version of the manuscript when assessing the changes.
>
> We would also like to clarify one point from our previous response that may have caused ambiguity. About dynamic weighting: Concretely, the system first identifies what the query is asking about, such as datasets, methods, compute resources, or limitations. It then selects the ResearchDB fields relevant to that intent. For example, a dataset query usually activates dataset and metric fields, while a method-evolution query activates method and architecture fields. The field weights are not learned by a new model; they are cold-start reference priors summarized from typical query patterns and pilot retrieval experiments. During retrieval, these priors are applied to the candidate fields selected by intent recognition; if only part of a prior applies, the remaining selected fields are re-normalized. Thus, the weights depend on the query intent but remain transparent and reproducible.
>
> To avoid overstatement, the updated manuscript describes this component as **rule-prior-initialized, intent-guided adaptive field weighting**. This wording is meant to clarify that the module is neither a fixed one-size-fits-all routing table nor an unconstrained learned weighting model; it is an inspectable query-planning step built on intent recognition and field-specific retrieval priors.

---

### Review · Reviewer_etyJ · 2026-05-28

**Summary Of Contributions:**

With the rapid development of AI research and the increasing number of publications in recent years, this paper proposes ResearchDB, a large-scale database, together with related data analysis frameworks. By combining topic clustering, large language model-assisted parsing, and structured retrieval, the authors provide a comprehensive representation of research activity, which supports the study of topic lifecycles, methodological transitions, dataset and model usage patterns, and institutional research directions. This is important for understanding the progress of rapidly developing AI research areas.

**Strength:**
1. The motivation is sound. Traditioinal bibliometric tools like citation counts and keywords miss much semantic information inside papers. Extracting methods, datasets, metrics, training settings, and GPU information is significant for experiment-heavy research subjects like AI.

2. ResearchDB is a database with usable structured fields, retrieval code and analysis scripts. It would be a useful community resource to understand the progress of AI research by providing a structured profiling of recent AI literature.

**Audience:**

Yes

**Audience Explanation:**

Against the backdrop of rapid advances in AI research and the increasing volume of scholarly publications, this study proposes an improved framework for organizing the literature. By enabling key experimental factors, including evaluation metrics, datasets, and hardware configurations, to be incorporated into the retrieval process, it offers valuable support for the broader research community.

**Claims And Evidence:**

Yes

**Claims Explanation:**

The claims in the paper are supported by extensive experiments.

**Requested Changes:**

1. Provide a small demo or a lightweight interface for querying ResearchDB. It would make the contribution easier to inspect and use.
2. Include qualitative error examples which demonstrate typical successful and failed extractions for methods, datasets, metrics and GPU information. It would help to understand practical reliability of the retrieval system.

---

> ### Author Response · Authors · 2026-05-30
> **Rebuttal for Reviewer etyJ.**
>
> We thank the reviewer for the positive assessment and constructive suggestions.
>
> **1. Lightweight demo / querying interface.**
>
> We agree that making ResearchDB easier to inspect and use is important. In the revision, we will clearly state that we will open-source the full codebase on GitHub. The released code will include the complete pipeline for building, querying, and analyzing ResearchDB, together with runnable lightweight demo scripts. These scripts will show how to query ResearchDB by fields such as conference, year, topic, dataset, method, metric, and institution, and how to return structured records, representative papers, and supporting evidence. We believe that releasing the full codebase, rather than only providing a limited online demo, will better support reproducibility, inspection, and further community development.
>
> **2. Qualitative error examples.**
>
> We agree that qualitative examples are helpful for understanding the practical reliability and limitations of ResearchDB. In the revision, we will add representative error examples in the supplementary material, covering methods, datasets, metrics, and GPU information extraction. These examples will illustrate typical failure cases such as overly generic method summaries, missing or confused dataset names, incomplete metric extraction, and missing GPU information when the original papers do not explicitly report hardware details. This will help readers better understand the current limitations and applicable scope of our extraction pipeline.

---

> ### Author Response · Authors · 2026-06-13
>
> We would like to add a brief follow-up to our previous response. We have uploaded an updated revision PDF, and we kindly ask the reviewer to refer to the latest version of the manuscript when assessing the changes.
>
> In the updated version, we added qualitative examples to make the reliability of ResearchDB easier to inspect. The successful case is illustrated by the retrieval and evidence-grounded synthesis example in Fig. A2. We also added a representative failure-case figure covering method and dataset extraction. These examples are intended to show both typical successful use and practical failure modes, such as overly generic method summaries and incomplete dataset coverage when many experimental settings are reported.

---

### Review · Reviewer_c56C · 2026-06-06

**Summary Of Contributions:**

The paper proposes a multidimensional, semantically driven framework for analyzing a large corpus of scientific publications. Its backbone is ResearchDB, which serves as the base for all subsequent trend analysis.
Main contributions:
(1) a large-scale profiling pipeline over 100K+ papers from 22 major conferences;
(2) field extraction via custom prompt-driven LLM parsing, with topic-keyword assignment combining clustering and LLM, followed by hierarchical retrieval; and
(3) use of this database for comprehensive trend analysis.

Strengths:
(1) the corpus is elaborate and covers most of the recent research landscape;
(2) the trend analyses are relevant and comprehensive, showing the value of this kind of study.

Weaknesses:
(1) in places the paper overclaims, presenting the framework as superior when the work is more accurately complementary  it leads on some dimensions while other techniques lead on others (e.g. Table 3, Fig. 9);
(2) the pipeline relies heavily on LLM parsing, a hallucination bottleneck some fields are validated but others are not, and the extraction limitations are underdiscussed. The role of clustering is unclear given the LLM already parses most complex fields directly.

**Audience:**

Yes

**Audience Explanation:**

Yes. Large-scale, semantically grounded analysis of AI research trends is of clear interest to the scientometrics and meta-research community, and the dataset/compute/institutional analyses offer findings useful for research planning, independent of the methodological concerns above.

**Broader Impact Concerns:**

No major ethical concerns. The work analyzes publicly available conference publications. One mild consideration worth a sentence in a Broader Impact note: the institutional and trend analyses are restricted to major English-language venues and depend on LLM extraction, so the resulting "research landscape" may under-represent non-English, regional, or lower-resourced research communities and could be read as more authoritative than the underlying coverage supports. Acknowledging this scope limitation would be sufficient, which the authors have already briefly touched upon in the limitation section.

**Claims And Evidence:**

Yes

**Claims Explanation:**

The high-level findings are plausible and the analyses are interesting, but several core claims are not yet adequately supported.
(1) The rationale for clustering over LLM-based topic classification rests on an untested empirical claim, with no head-to-head comparison. (2) The "dynamic" field weighting is in fact preset rule-based routing.
(3) ResearchDB's schema is specified inconsistently across the paper, undermining the "transparent/reproducible" framing.
(4) The compute trend may conflate rising demand with rising reporting rates, and rests on the lowest-accuracy extracted field.
(5) The claim that the framework is best "across all metrics" is contradicted by the paper's own Table 3 and Fig. These are addressable, primarily through added comparisons/ablations, accurate descriptions, and a single authoritative schema specification.

**Requested Changes:**

### Critical

1. **(Section 3.1)** The justification for using clustering rather than LLMs for topic classification rests on the claim that LLMs hallucinate and cannot stably output classifications at scale. But the pipeline already parses every paper with an LLM and uses ChatGPT-5 for topic summarization and naming, so per-paper LLM compute is already spent it is unclear why classification needs a separate clustering stage rather than reusing that pass. The claim is empirical but untested; no comparison is given against a direct LLM-based classification baseline. Please add such a comparison on a labeled subset; if clustering is better, the scores should show it.
2. **(Section 4)** The field weighting is described as "dynamic," but Appendix B shows preset static rules mapping query categories to fixed weight vectors (e.g. dataset queries → datasets 0.35, metrics 0.25, ...). The LLM selects among preset rules rather than generating weights freely, so "dynamic" is misleading the mechanism is closer to rule-based routing. Please describe it accurately, and note that no ablation establishes its benefit over uniform/fixed weighting.
3. **(Section 4.2 / Table A6)** Table A6 reports compute-field accuracy but not coverage. Please report the fraction of papers that actually contain compute information and how the pipeline handles papers with none (empty vs. inferred). More importantly, compute reporting has increased over time, so the temporal trend in Fig. 5 (left) may conflate rising compute demand with rising reporting rates especially for 2020–2022 papers. Please address whether the trend controls for reporting-rate changes; as presented it cannot distinguish the two.
4. **(Appendix B)** The fields defining ResearchDB are inconsistent across the paper: the extraction prompt outputs 21 fields, the searchable fields list a smaller subset (~13), and Section 3.2 refers to "core structural fields" (abstract, methods, datasets, limitations) as a third grouping, with no explanation of how these relate. Field names also differ (`keywords_description` in Table 2 vs `keywords_explanation` in the prompt). For an artifact positioned as transparent and reproducible, a single authoritative schema specification is needed ideally with a worked example tracing one query through extraction, storage, and retrieval. As presented, the actual structure of ResearchDB cannot be determined.
5. **(Section 5.1 / Table 3)** The claim that "the full framework consistently achieves the highest performance across all metrics" is contradicted by Table 3, where the w/o-MF and SF variants score higher on Readability. Please revise to reflect the results accurately (e.g. ResearchDB leads on N of 5 dimensions, with SF best on Readability). Reporting an aggregate score (mean across dimensions) for all methods, alongside the per-dimension scores, would also make the comparison clearer.


### Strengthening

1. **(Table 2)** The descriptions for `keywords` / `keywords_description` and for `abstract_ori` / `abstract_summary` are identical. If these are distinct fields, give each a unique description; if redundant, simplify and explain. As written, a reader cannot tell what each stores.
2. **(Section 3.1)** The five dimensions (Summary, Technical, Analysis, System, Post-Processed) read as an exhaustive list rather than a high-level grouping. Please clarify they are broad categories containing sub-fields, and reflect this grouping in Table 2.
3. **(Section 3.2)** The pipeline decomposes the query into sub-questions and, separately, performs an intent-driven semantic search over the metadata-filtered subset. It is unclear why intent is handled in two stages please clarify what each contributes, or consolidate if redundant.
4. **(Section 4.1, Fig. 4)** The embedded time-series plot (lower left) for the Top-20 topics does not label which topics are shown. Please add labels or a reference table. One topic (shown as #19) has markedly higher citations than all others across every year it would benefit from being explicitly identified and briefly discussed if relevant to the lifecycle analysis.
5. **(Section 4.3, Fig. 6a)** The caption states the bar chart shows "annual paper counts for each dataset type," but no dataset-type breakdown is visible the chart appears to show overall dataset/benchmark counts per year. Please correct the caption or add the intended type dimension.
6. **(Section 4.3,Dataset landscape Fig. 7a)** The ImageNet trend plateaus rather than declines significantly, so "saturation" overstates it. The ImageNet-vs-COCO cross-modal contrast is reasonable as a speculative direction but should be framed as such, not as a definite conclusion.
7. **(Section 4.4, Fig. 7b)** The figure is hard to read as a direct comparison across many institutions. Consider highlighting a few representative cases (e.g. two leading academic, two leading industrial) for legibility, with the full set in an appendix.
8. **(Section 5, Fig. 9)** The radar chart uses a 0–5 scale but does not print exact per-dimension scores. Reporting the numeric values (e.g. a table), ideally with variance, would let readers judge whether differences between ResearchDB, SurveyX, and ChatGPT-5 are meaningful rather than within annotator noise.
9. **(Section 5)** Cohen's kappa is defined for two raters, but agreement is reported across five annotators. Please clarify whether Fleiss' kappa (or Krippendorff's alpha) was intended, and report the value.
10. **(Section 5, Fig. 9)** The figure suggests SurveyX matches or exceeds ResearchDB on Coverage. Any overall-superiority claim should be scoped to the dimensions where the advantage holds.
11. **(Appendix B)** The weights are applied to similarity scores over "core structural fields  abstract, methods, datasets, limitations." What exactly are the core structural fields, and does retrieval match the query against only these or against all ResearchDB schema fields?
12. **(Appendix D, Table A6)** The manual evaluation of semantic fields uses five annotators but reports no inter-annotator agreement. Since the summarization evaluation does report agreement, adding a kappa/alpha here would similarly substantiate the extraction-score reliability.
13. **(Section 6)** The limitations mention "potential LLM parsing biases" only generically. Please acknowledge the specific, quantified limitations the ~70% compute-field extraction accuracy and the temporal reporting-rate bias affecting the compute trend since these directly bear on the reliability of those analyses.

---

> ### Author Response · Authors · 2026-06-13
>
> We thank the reviewer for the careful comments and positive assessment of the corpus and trend analyses. We respond to the reviewers' concerns as follows:
>
> **1. Clustering and LLM Topic Classification**
>
> Because our corpus is open-ended and spans years and venues, independent per-paper LLM labels may introduce inconsistent names and drifting granularity. We use BERTopic-style clustering to form stable semantic neighborhoods, and use ChatGPT-5 only to summarize, name, and refine cluster hierarchies. While an LLM-only comparison would be valuable, the revised paper removes the superiority claim and scopes clustering to stable topic structures for lifecycle analysis.
>
> **2. Intent-Guided Field Weighting**
>
> We thank the reviewer for pointing out the ambiguity in “dynamic weighting.” We now clarify that the mechanism is neither a fixed routing table nor a learned weighting model. It is rule-prior-initialized, intent-guided adaptive field weighting. The LLM-based intent prompt first selects relevant candidate fields from the ResearchDB schema; different queries therefore activate different field combinations.
> After candidate fields are selected, weights are initialized using Appendix B priors summarized from typical query patterns and pilot retrieval experiments. They provide an interpretable starting point but are not applied identically to all queries. We also clarify the ablation: w/o IFW keeps the same selected fields but replaces intent-guided weights with uniform weights.
>
> **3. Compute Trend and Reporting Bias**
>
> We agree that compute trends require stronger qualification. The revised manuscript states that papers without GPU or training-resource information are stored as empty/unknown and are not inferred. Thus, Fig. 5 reflects reported compute usage among papers that explicitly disclose compute resources, not absolute compute demand across the whole corpus. We also emphasize that gpu_info is a lower-reliability field because many papers omit hardware, accelerator count, or training time. To avoid overstating this analysis, we do not add a reporting-rate-corrected growth claim; instead, we scope the conclusion to reported compute usage and list reporting bias as a limitation.
>
> **4. ResearchDB Schema**
>
> We agree that the original schema description was inconsistent. The revised paper separates high-level field groups from the complete field-level schema. Table 2 summarizes five groups: Info, Summary, Technical, Analysis, and System. Appendix A provides the authoritative schema with field name, group, origin, and purpose, used consistently across extraction, storage, retrieval, and analysis.
> We fix field-name inconsistencies by consistently using keywords_description for LLM-generated keyword explanations. We distinguish keywords from keywords_description, and abstract_ori from abstract_summary. We remove the ambiguous “core structural fields” wording and add a worked example covering filtering, field selection, weighting, retrieval, and evidence-grounded summarization.
>
> **5.  Fig. 9, and Overclaims**
>
> We agree that the previous “best across all metrics” claim was inaccurate. We remove it and revise the discussion to match the results. The ablation table now reports mean scores: ResearchDB has the highest mean and leads in accuracy, novelty, and usefulness.We also correct the SurveyX/ChatGPT-5 comparison: ResearchDB leads on accuracy, novelty, and usefulness; SurveyX on coverage; and ChatGPT-5 on readability.
>
> **6. Other Revisions**
>
> We clarify query decomposition versus intent-guided retrieval, connect the Top-20 topic inset to Appendix C, identify topic 19, correct the Fig. 6a caption, soften the ImageNet claim to a plateauing-benchmark observation, and frame the ImageNet-COCO contrast as suggestive. For the five-annotator evaluation, the Fleiss’ kappa value is 0.71, indicating substantial agreement. We also add a Data and Code Availability statement and qualitative extraction examples, including representative failure modes for methods, datasets, metrics, and GPU information.
>
> Finally, we strengthen limitations and broader impact: the corpus is restricted to major English-language AI venues and may under-represent non-English, regional, or lower-resourced communities. We also note that automated trend analysis may create popularity feedback loops if frequently retrieved or fast-growing topics are interpreted as inherently more important. ResearchDB is framed as an exploratory, evidence-grounded analysis tool, not an authoritative global ranking.

---

> > ### Comment · Reviewer_c56C · 2026-06-18
> >
> > I thank the authors for the thorough and responsive revision. I have reviewed the updated manuscript and confirm that my main concerns: the clustering claim, compute framing and reporting-bias limitation, schema reorganization, the IFW ablation, and the corrected performance claims are addressed.
> > One minor point I could not locate: the Fig. 7b institutional-comparison readability suggestion (highlighting a few representative academic/industrial cases for legibility). If this was addressed, a pointer would be appreciated; if not, it is a minor presentation item and not critical to my recommendation.
> > I appreciate the careful engagement with the review.